# Atmospheric boundary layer dynamics from balloon soundings worldwide: CLASS4GL v1.0

Hendrik Wouters[1], Irina Y. Petrova[1], Chiel C. van Heerwaarden[2], Jordi Vilà-Guerau de Arellano[2], Adriaan J. Teuling[2], Vicky Meulenberg[2], Joseph A. Santanello[3], and Diego G. Miralles[1]

[1]Ghent University, Laboratory of Hydrology and Water Management, Coupure Links 653, 9000 Ghent, Belgium
[2]Wageningen University and Research, Dept. of Environmental sciences, Postbox 47, 6700AA Wageningen, The Netherlands
[3]National Aeronautics and Space Administration, Goddard Space Flight Center, Hydrological Sciences Laboratory (617), Greenbelt, MD, USA

**Correspondence:** Hendrik Wouters (hendrik.wouters@ugent.be)

**Abstract.** The coupling between soil, vegetation and atmosphere is thought to be crucial in the development and intensification of weather extremes, especially meteorological droughts, heatwaves and severe storms. Therefore, understanding the evolution of the atmospheric boundary layer (ABL) and the role of land–atmosphere feedbacks is necessary for earlier warnings, better climate projection and timely societal adaptation. However, this understanding is hampered by the difficulties to attribute cause–effect relationships from complex coupled models, and the irregular space–time distribution of *in situ* observations of the land–atmosphere system. As such, there is a need for simple deterministic appraisals that systematically discriminate land–atmosphere interactions from observed weather phenomena over large domains and climatological time spans. Here, we present a new interactive data platform to study the behaviour of the ABL and land–atmosphere interactions based on worldwide weather balloon soundings and an ABL model. This software tool – referred to as CLASS4GL (http://class4gl.eu) – is developed with the objectives to (a) mine appropriate global observational data from ∼ 15 million weather balloon soundings since 1981 and combine them with satellite and reanalysis data, and (b) constrain and initialize a numerical model of the daytime evolution of the ABL that serves as a tool to interpret these observations mechanistically and deterministically. As a result, it fully automizes extensive global model experiments to assess the effects of land and atmospheric conditions on the ABL evolution as observed in different climate regions around the world. The suitability of the set of observations, model formulations and global parameters employed by CLASS4GL is extensively validated. In most cases, the framework is able to realistically reproduce the observed daytime response of the mixed-layer height, potential temperature and specific humidity from the balloon soundings. In this extensive global validation exercise, a bias of 10.1 m h$^{-1}$, $-0.036$ K h$^{-1}$ and 0.06 g kg$^{-1}$ h$^{-1}$ is found for the morning-to-afternoon evolution of the mixed-layer height, potential temperature and specific humidity. The virtual tool is in continuous development, and aims to foster a better process-understanding of the drivers of the ABL evolution and their global distribution, particularly during the onset and amplification of weather extremes. Finally, it can also be used to scrutinize the representation of land–atmosphere feedbacks and ABL dynamics in Earth system models, numerical weather prediction models, atmospheric reanalysis, and satellite retrievals, with the ultimate goal to improve local climate projections, provide earlier warning of extreme weather, and foster a more effective development of climate adaptation strategies. The tool can be easily downloaded via http://class4gl.eu and is open source.

# 1 Introduction

Climate and weather phenomena are largely influenced by land surface processes and the characteristics of the landscape. The interactions between soil, vegetation and atmosphere are thought to be particularly important for the evolution of extreme weather events such as droughts, heatwaves and convective thunderstorms (Santanello et al., 2018; Miralles et al., 2018; Seneviratne et al., 2010). The quantification of the drivers behind the extreme events is challenging, yet an understanding of the physical mechanisms underlying these events is highly relevant for earlier societal warning, better climate projection and timely adaptation (Sillmann et al., 2017). First efforts to quantify the relevance of land–atmosphere feedbacks date back to the late 20th century (e.g., Ek and Mahrt, 1994; Betts and Ball, 1995). However, substantial advancements have occurred in recent years after climate modelling initiatives such as the Global Land–Atmosphere Coupling Experiment (GLACE Koster et al., 2006; Guo et al., 2006; Berg et al., 2015), and observation-based studies under the umbrella of the Global Energy and Water Exchanges (GEWEX) Local Land–Atmosphere Coupling (LoCo) project (Roundy et al., 2013; Santanello et al., 2015; Tawfik et al., 2015; Santanello et al., 2018). Recent studies have highlighted the importance of soil moisture and evaporation for the occurrence of afternoon rainstorms (Findell et al., 2011; Taylor et al., 2012; Guillod et al., 2015; Petrova et al., 2018), droughts (Roundy and Santanello, 2017; Teuling et al., 2013) and extreme heat events (Fischer et al., 2007; Miralles et al., 2014). Moreover, studies on anthropogenic land-cover change and land management — such as deforestation (Akkermans et al., 2013; Lejeune et al., 2014), irrigation (Thiery et al., 2017; Lawston et al., 2015), modified croplands (Seneviratne et al., 2010) and urban expansion (Wouters et al., 2017) — have demonstrated profound influences of land conditions on local and regional climate, and specifically on the occurrence of extreme weather.

However, assessing cause–effect relationships in observational and model studies of land–atmosphere interactions remains complex, given the cross-correlation of multiple climate variables without the need of implying causation, the bidirectional interactions within the system, the various scales of variability and autocorrelation of different elements, and the unavoidable confounding effect of unobserved causal variables (Miralles et al., 2018). Likewise, the many studies of land–atmosphere interactions based on the use of global or regional climate models are model dependent and only poorly constrained by observations (Orlowsky and Seneviratne, 2010; Davin et al., 2019), and the complexity of the Earth Systems Models hampers the assessment of individual feedback processes. An intermediate compromise between statistical analysis of observations and complex climate model simulations could close this gap in process-understanding. For example, mechanistic studies based on simpler models supported by observations have yielded new insights recently (Roundy et al., 2013; Zaitchik et al., 2013; Santanello et al., 2009). Particularly, atmospheric boundary layer (ABL) models have been initialized and constrained with observations to simulate the atmospheric response to land surface conditions and the state of the free atmosphere; this way, the influence of turbulent heat fluxes, incoming radiation, subsidence, advection, or entrainment can be easily quantified within certain ranges of uncertainty (Pietersen et al., 2015; Miralles et al., 2014; Ouwersloot et al., 2012; van Heerwaarden et al., 2009). For example, ABL models have been applied to investigate soil moisture and vegetation feedbacks during combined droughts and heatwaves in Europe (Miralles et al., 2014), the different feedbacks on heatwave evolution over forests and grasslands (van Heerwaarden and Teuling, 2014), and the suppression of clouds by plants in a $CO_2$-rich atmosphere (Vilà-Guerau de Arellano

et al., 2012). The advantage of using these ABL bulk models is twofold: (a) unlike climate models, they can be routinely initialized and constrained by observations and easily interpretable in terms of the interaction between variables; (b) unlike merely statistical analysis of observational data, they provide unambiguous understanding on the deterministic links among the variables in the system. Yet, these mechanistic models require detailed observations describing the entire state or evolution of the soil, vegetation and atmosphere. Given this dependency, the previously-mentioned process-based studies usually focus on one particular location only, for one or a few diurnal cycles at best. A generalization of the mechanistic outcomes of these process-based case studies to other climate regions or periods of extreme events may not always be justified. By all means, the atmospheric and land surface observations are irregular in time and space, which makes it very challenging to attribute the causes of meteorological variability.

Here, an open source interactive data platform is presented based on the application of the Chemistry Land-surface Atmosphere Soil Slab model (CLASS; Vilà-Guerau de Arellano et al., 2015; van Heerwaarden et al., 2010) to balloon soundings worldwide. The platform, hereafter referred to as the CLASS4GL (CLASS model for GLobal studies), is designed to mine observations from the radio soundings, satellite remote sensing observations, reanalysis data and surface data inventories to constrain and initialize the ABL model (CLASS). It automizes mass parallel simulations of the ABL and enables global sensitivity experiments. As a result, it is designed for studies that aim to foster a better understanding of the dynamics of ABL and the development of extreme weather, and allows the attribution of changes in the state of the ABL to specific land and atmospheric conditions. A core goal of this study is to present this interactive data platform, including a summary description of the ABL model and the data mining procedure used to initialize and constrain model simulations (Sect. 2). Furthermore, the skill of the modelling framework to reproduce the daytime evolution of the ABL is evaluated against worldwide observations from specific field campaigns as well as operational balloon soundings (Sect. 3). Finally, a perspective is provided in which the potential of this framework to contribute to a better understanding of land–atmosphere interactions over different climates is discussed (Sect. 4).

## 2  Data and methods

The CLASS4GL platform is composed of three modules (see Fig. 1), namely an ABL model (CLASS), a data mining module, and an interface module. The ABL model is used simulate the ABL evolution and is described in detail under Sect. 2.1. It requires appropriate observations of the ABL for the initialization in the morning and for the validation in the afternoon. Meanwhile, the data mining module collects profile observations from soundings taken during research campaigns and operational activities since 1981. The intensive research campaigns offer continuous high-quality sounding profiles available for specific time periods and locations, while the operational weather balloon soundings offer more regular balloon launches at a vast amount of locations around the world, but with intermittent and varying quality. The profile database is extensive, yet spatially and temporally sparse. Quality check tools have been applied to mine the profiles that are appropriate for the ABL model in the way described in Sect. 2.2. In order to further constrain and initialize ABL model simulations with surface conditions and larger-scale atmospheric variables, the data mining module also employs ancillary data from satellite remote

sensing, reanalysis and survey inventories (Sect. 2.3). File formats are NetCDF and YaML to enable the easy adoption of any (upcoming) input datasets and the exchange of profiles, parameters and model experiments among users and a central database. Finally, the interface module provides the ability to easily perform multiple simulations of the diurnal ABL evolution in parallel, as well as batches of sensitivity experiments. Therefore, it enables the parallelization of multiple model simulations and

5    offers multi-processing support for both regular computer workstations as well as super-computing infrastructure. The interface module also implements a range of tools for pre-and post-processing the sparse data pool of inputs and experiments, and a data explorer with a graphical user interface. As a result, CLASS4GL automizes mechanistic assessments of the observed diurnal ABL behaviour around the globe, and allows the exploration of local land–atmosphere feedbacks and the attribution of cause–effect relationships. A detailed description of the platform is provided in the next subsections. CLASS4GL is provided

10   as an open source Python library, it is conveyed under the GNU General Public License version 3 (GPLv3), and it can be easily downloaded via http://class4gl.eu.

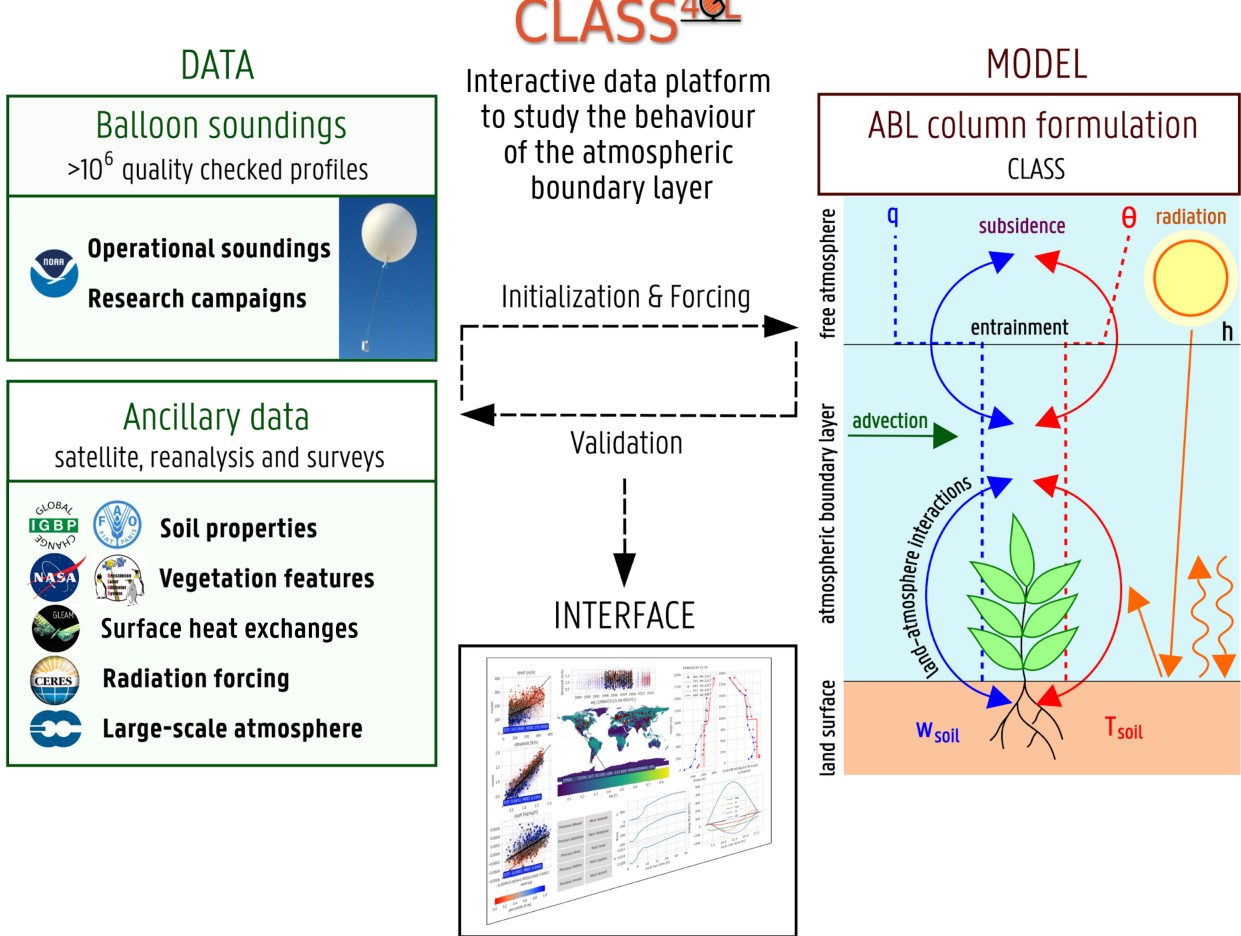

**Figure 1.** Schematic overview of the Chemistry Land-surface Atmosphere Soil Slab model for GLobal studies (CLASS4GL)

## 2.1 ABL model: CLASS

The Chemistry Land-surface Atmosphere Soil Slab model (CLASS) is a conceptual bulk model that uses a small set of differential equations to reproduce the evolution of the ABL's essential properties over diurnal time spans in response to surface and atmospheric forcings and feedbacks. The reason of choosing a bulk model is the low computational cost and the easy interpretation of the canonical ABL properties in view of extensive global experiments. However, we note that other models that allow for a more complex and explicit representation of the turbulent processes in the ABL – such as single column ABL models or large-eddy simulation models – may also be implemented.

CLASS is based on the original work by Tennekes (1973) and uses the thermodynamic equations of the ABL proposed by Tennekes and Driedonks (1981). The ABL is represented as a single model layer. The use of the mixed-layer equations implies that the turbulence and vertical gradients inside the mixed layer are not explicitly resolved, and the potential temperature ($\theta$), specific humidity ($q$) and wind components are assumed to be homogeneous within the mixed layer. This assumption tends to be supported by the efficient turbulent mixing under convective conditions (Bauer, 1908). At the top of the mixed layer, the entrainment of heat and moisture is parameterized by a jump of $\theta$, $q$ and wind components over an infinitesimally small height, which are initialized with a constant lapse rate with height in the overlying free atmosphere. Entrainment flux is calculated as a fixed fraction (0.2) of the buoyancy flux, to which one also adds the entrainment flux driven by shear. An important feature of the model is the possibility to represent the subsidence coupled to the entrainment process at the inversion zone (Vilà-Guerau de Arellano et al., 2015). Herein, the subsidence velocity is a function of the divergence of the mean horizontal wind and the evolving mixed-layer height. The surface–atmosphere turbulent exchanges for momentum, heat and moisture in the surface layer are calculated considering their aerodynamic resistances. These are calculated in an iterative way assuming constant values for aerodynamic roughness lengths, while applying correction factors for non-neutral stratification of the atmospheric surface-layer (Paulson, 1970) according to the Monin–Obukhov similarity theory (Monin and Obukhov, 1954). It should be kept in mind that more realistic profiles with explicit ABL gradients for temperature, humidity and wind speed – especially at the top (entrainment zone) and bottom (surface layer) of the mixed layer – are not yet considered by the model. In order to tackle these limitations and associated uncertainties, more research is needed employing more realistic profiles. The surface energy balance at the land surface is solved using the Penman–Monteith equation (Monteith, 1965), and the heat and moisture transport in the soil is described using a two-layer force–restore model (Noilhan and Planton, 1989; Noilhan and Mahfouf, 1996) employing empirical relations for soil hydraulic properties of (Clapp and Hornberger, 1978). The amount of net radiation available for the sensible, latent and ground heat flux is calculated from the albedo, emissivity the incoming shortwave and long-wave radiation, the surface temperature and the cloud cover. For the vegetation fraction, the transpiration response of vegetation to the atmospheric conditions considers the empirical stomatal resistance from Jarvis (1976) inversely proportional to the leaf area index and the plant water stress — assumed to be a linear function of the soil-moisture deficit of the deep soil layer — and also a function of the incoming shortwave radiation, the vapor pressure deficit and the air temperature. An analogous formulation is used for the resistance of the bare-soil fraction, but considering it only as an inverse linear function to the soil-moisture deficit of the shallow soil layer. More details and explicit formulations of the coupled land–atmosphere system

CLASS can be found in Vilà-Guerau de Arellano et al. (2015), van Heerwaarden and Teuling (2014) and van Heerwaarden et al. (2010).

In order to exploit the full database in CLASS4GL, the ABL model is upgraded with the following features:

- A representation of advection as an additional atmospheric dynamic forcing (see below).

- A representation of the upper-air atmospheric profile that also evolves according to the external large-scale dynamic forcing of advection and subsidence, and as such accounts for varying (instead of constant) lapse rates of the capping inversion during the growth of the mixed layer.

- An implementation of alternative surface-layer transfer coefficients for momentum and heat Wouters et al. (2012), which account for additional non-neutral stability correction factors for the roughness sublayer (Ridder, 2009). The procedure uses a non-iterative approximation of the transcendental relation between the transfer coefficients and the Richardson bulk number, hence preserves numerical stability and decreases the computational cost of CLASS.

- An iterative procedure to invert soil moisture conditions based on the match of simulated evaporative fraction (*EF*) to satellite-based *EF* estimates. Details on this procedure can be found under Sect. 2.3 (eq. 3).

To conclude this section, we elaborate the above-mentioned CLASS model extension to consider advection as additional ABL dynamic forcing. It is assumed that both the mean vertical wind speed and also horizontal turbulent fluxes in the mixed layer are negligible, hence set to zero. As such, the laws of conservation of momentum, energy, and atmospheric constituents in the mixed layer are given as follows:

$$\frac{\partial \overline{\psi}}{\partial t} + \overline{u}\frac{\partial \overline{\psi}}{\partial x} + \overline{v}\frac{\partial \overline{\psi}}{\partial y} + \frac{\partial \overline{w'\psi'}}{\partial z} = \overline{S}_\psi \tag{1}$$

where $\psi$ is a generic variable ($\theta$, $q$, $u$, $v$), $u$, $v$, $w$ are the wind fields, the overbars indicate the Reynolds averages, $\psi'$ indicates the fluctuation around the averages (note the difference from eq. 2.5 of Vilà-Guerau de Arellano et al. (2015) because of the horizontal advection terms), and $\overline{S}_\psi$ represents the sum of the external forcings, sources and sinks of $\psi$. Integrating the last equation over the whole mixed-layer column in both the x, y and z direction yields the tendency of the bulk mixed-layer quantity (assuming no forcings, sources or sinks in the mixed layer):

$$\frac{\partial \langle \psi \rangle}{\partial t} = \frac{1}{h}\left[\langle (w'\psi')_s \rangle - \langle (w'\psi')_e \rangle\right] - \langle \overline{\mathbf{U}} \cdot \nabla_{\mathrm{hor}} \overline{\psi} \rangle \tag{2}$$

where $(w'\psi')_s$ is the vertical turbulent flux at the surface, and $(w'\psi')_e$ is the entrainment flux at the mixed-layer height ($h$), $l$ is the horizontal extent of the single column, $\langle \rangle$ indicates the bulk (slab or mixed-layer) mean value of any field over the entire mixed-layer column, and $\langle -\overline{\mathbf{U}} \cdot \nabla_{\mathrm{hor}} \overline{\psi} \rangle$ is the bulk horizontal advection for $\overline{\psi}$ in the mixed layer, which is obtained from external forcing fields.

## 2.2 Automized balloon data mining

Global data of weather balloon soundings are taken from the Integrated Global Radiosonde Archive (IGRA; Durre et al., 2006) which is maintained under the auspices of the National Oceanic and Atmospheric Administration (NOAA). The IGRA archive

is routinely updated and currently includes more than 2700 stations covering major global climate regions. The CLASS4GL sounding database is additionally supplemented with data from intensive radiosonde campaigns from HUMPPA (Williams et al., 2011), BLLAST (Pietersen et al., 2015) and GOAMAZON (Martin et al., 2016) – see Tab. 1 and Fig. 2. Other sources of vertical profile data (from e.g., aircraft, satellites, other observation campaigns or long-term operational soundings) may

5    be considered in future applications of the framework. As described above, CLASS requires morning sounding profiles for initialization and afternoon profiles for validation to enable a mechanistic interpretation of the diurnal ABL evolution.

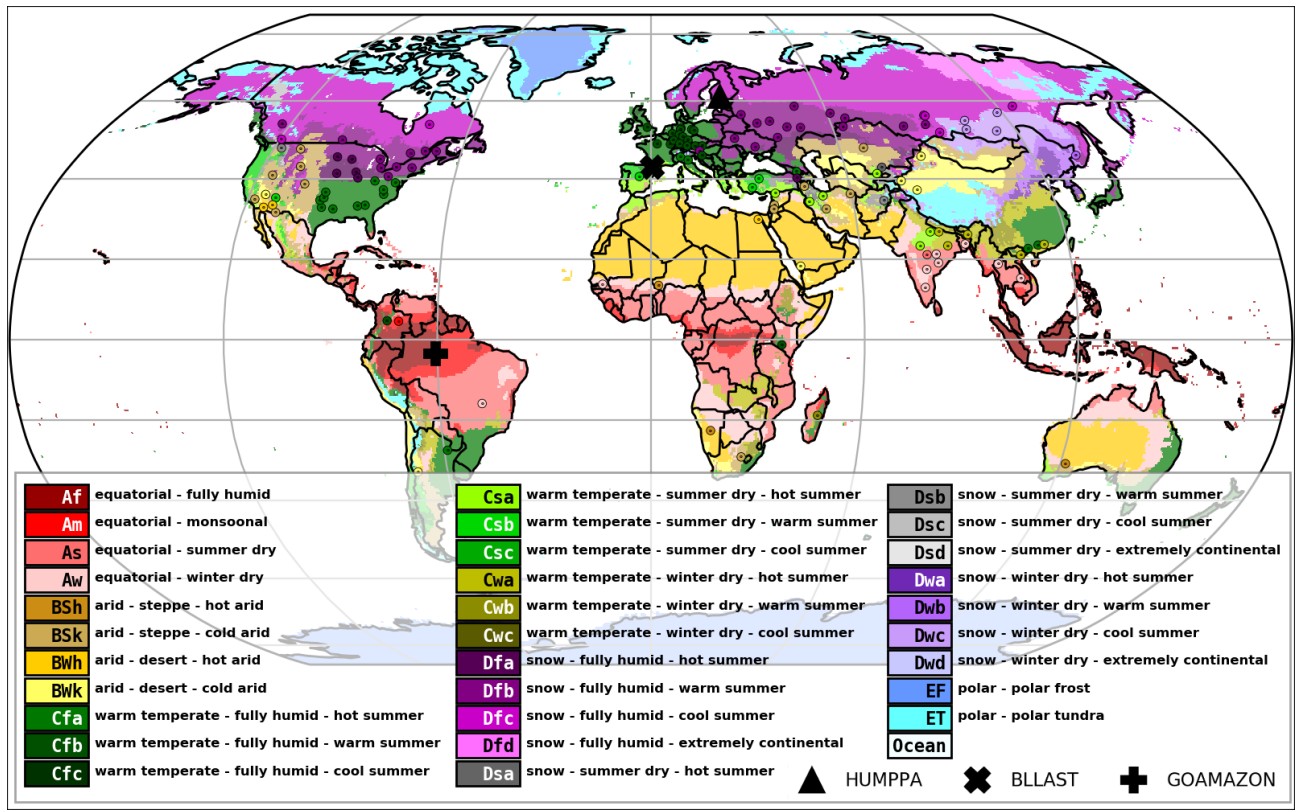

**Figure 2.** Distribution of the launch sites from the Integrated Global Radiosonde Archive (IGRA) retained after the profile quality selection procedure. The different climate classes are indicated with the colors according to the Köppen–Geiger climate classification. The markers indicate the locations of the three observation campaigns (i.e., HUMPPA, BLLAST and GOAMAZON). The profile quality selection is described in presented in Sect. 2.2.

All balloon sounding profiles (∼15 million profiles) are pre-processed first by calculating the bulk mixed-layer properties: The mixed-layer height ($h$) is assessed as the height at which the Bulk Richardson number ($RiB$) exceeds a critical value ($RiB_c$). We adopt the estimates for $RiB_c$ provided by Zhang et al. (2014): $RiB_c = 0.24$ for strongly stable boundary layers,

10    $RiB_c = 0.31$ for weakly stable boundary layers, and $RiB_c = 0.39$ for unstable boundary layers. The uncertainty range of $h$ (used below) is determined from its interval corresponding to the $RiB_c$ range $[0.24, 0.39]$, for which the interval is further

extended to the nearest sounding records above and below. Second, the mixed-layer potential temperature ($\theta$), specific humidity ($q$), zonal wind ($u$) and meridional wind ($v$) are calculated as their average values recorded within the mixed layer. The capping inversion is estimated by a linear extrapolation of the two lowest sounding measurements above h, for which its lapse rate for potential temperature ($\gamma_\theta = d\theta/dz$), specific humidity ($\gamma_q = dq/dz$) and wind components ($\gamma_u = du/dz$ and $\gamma_v = dv/dz$) are

calculated. The jump values at the $h$ for potential temperature ($\Delta\theta$), specific humidity ($\Delta q$) and wind components ($\Delta u$ and $\Delta v$) are estimated from the difference between the values of the capping inversion at $h$ and the values within the mixed layer.

  Afterwards, morning–afternoon profiles are selected that meet a series of selection criteria: the morning profiles, ie. profiles before 12 h local time, are selected first, and they amount to ∼6 million profiles. Here, the selection of suitable morning soundings (and the subsequent afternoon soundings after 12 h is based on the timing of these soundings (a): Morning (and

afternoon) sounding profiles ideally should be acquired after sunrise and before sunset, respectively. However, routine sounding launches happen synchronously on a daily basis at 0 h and 12 h UTC, whereas launches at intermediate timings (3 h, 6 h, 9 h, 15 h and 18 h UTC) are rare. As a result, many launches, especially those at 0 h UTC in Europe and Africa, often happen several hours before sunrise. Since the net exchanges near the surface for heat, moisture and radiation are generally low at the end of the night, the atmospheric profiles tend not to change dramatically before sunrise (unless the synoptic situation

changes), being often representative for the time the mixed layer starts to emerge (van Stratum and Stevens, 2018). As such, in order to maintain a high number of soundings in our analyses, launch times within 3 h prior to sunrise are still allowed here. For these soundings, the ABL simulation starts at sunrise, assuming that the change in the atmospheric profile since the balloon launch time is negligible. Furthermore, (b) only those soundings are retained with more than seven measurements in the vertical below 3000 m (72% of the morning soundings), (c) for which the uncertainty of the mixed-layer height is lower

than 150 m (26% of the morning soundings), (d) for which a well-mixed layer is observed (ie. for which the root-mean square deviation of the temperature from the estimated mixed-layer average is lower than 1.5°C; this criterion is met by 92% of the morning soundings). We also (e) set the morning lower temperature limit to 278 K in order to minimize the chance of freezing temperatures during the course of the simulations (this criterion is met by 70% of the morning soundings). The next criterion is that (f) an afternoon sounding can be found with the same criteria as the morning sounding except regarding the uncertainty of

the mixed-layer height (which is met by 24% of the filtered morning soundings). Here, the afternoon radiosonde profile on the same day needs to occur between local noon and 1 h before sunset (defined as the time when the incoming shortwave radiation at the top of the atmosphere becomes zero), and at least 4 h after the model initialization by the morning sounding so that a sufficiently large model time span is considered. In case there are more than two soundings retained during a particular day which especially occurs during the campaigns, the sounding closest to sunrise is taken for the initialization in the morning, and

the latest sounding for the validation in the afternoon. so that a sufficiently large model time span is considered. We also require that (g) all external forcing parameters described in the next Section are available for the simulation (which is met by 8.7% of the filtered sounding pairs). The above criteria lead to 21,826 profile pairs from 134 stations. Finally, the current version of CLASS is only capable of representing growing mixed layers. Therefore, an observed mixed-layer growth of 40 m h⁻¹ is considered as a lower limit (which is met by 85% of the profile pairs), which leads to 18,385 profile pairs from 121 stations.

For the three intensive observation campaigns, 22 out of 49 profiles are retained. An overview of the global distribution of the

retained radiosondes and their corresponding climate regime is shown in Fig. 2. These requirements are flexible and may be reconsidered according to the intended application, since there is an obvious trade-off between sounding quality and amount of data being retained.

It should be noted that many sites have only a few days with sounding pairs that meet the selection criteria, and most often, there are only intermittent time series available. For applications that require continuous datasets, the extraction profiles from reanalysis data, satellite-based products, or even Earth System Models, is also implemented in CLASS4GL. This alternative to the use of sounding data holds great promise for spatially-explicit climatological ABL studies and multi-annual trend assessments. ABL model simulations using continuous/gap-filled datasets may deviate from those using the observations. Hence, an additional validation is needed as soon as such datasets are employed, in which should compare the gap-filled datasets with the observations and the error propagation on the ABL model simulations. Such an in-depth evaluation against the available sounding pairs can be done using the present framework.

## 2.3 Gridded ancillary data

In addition to balloon observations, CLASS4GL uses gridded data of the land and large-scale atmospheric state to initialize and constrain ABL model simulations. These ancillary data aim at complementing the soundings and providing context regarding the land and atmospheric conditions for which the sounding measurements take place. In total, we use four satellite-based products, two survey datasets and one reanalysis to describe soil–vegetation conditions and large-scale atmospheric forcing. All input datasets and parameters are listed respectively in Tab. 1 and 2, and details can be found in the next paragraphs.

Static datasets are used to describe initial soil properties and land cover parameters, such as the fraction of land covered by vegetation and the surface albedo. The latter are based on the global vegetation continuous fields product from the Moderate Resolution Imaging Spectroradiometer (MODIS MOD44B; Hansen et al., 2005). Wilting point, soil porosity, field capacity and critical soil moisture are derived from the database of Global Gridded Surfaces of Selected Soil Characteristics from the International Geosphere-Biosphere Programme (IGBP-DIS, 2000). The Harmonized World Soil Database (HWSD; Nachtergaele et al., 2009) is used to provide soil classes. From this class, the Clapp and Hornberger parameters and the thermal parameters for the force–restore method are obtained via the look-up table in Noilhan and Planton (1989). The vegetation canopy height is determined from the Geoscience Laser Altimeter System (GLAS; Simard et al., 2011). Dynamic data of Leaf-Area Index (LAI) are taken from the Global Inventory Modeling and Mapping Studies (GIMMS) (Liu et al., 2012).

Initial surface and root-zone soil moisture values are inverted using the Global Land Evaporation Amsterdam Model (GLEAM) version 3.2a (Miralles et al., 2011; Martens et al., 2017) as reference. In order to maximize the consistency between CLASS and GLEAM, the soil and vegetation ancillary data used by CLASS4GL, and described above, correspond in fact to those used by the GLEAM v3.2a. It should be noted that CLASS and GLEAM have similar but not identical surface-vegetation-atmosphere transfer schemes, hence equivalent soil moisture levels may be associated to a differing *EF*. In order to minimize potential incompatibilities, CLASS4GL inverts soil moisture values by iteratively converging to the *EF* from GLEAM, instead of using the GLEAM root-zone soil moisture directly to initialize the ABL simulations. This iterative procedure is based on finding the

zero of the following function:

$$f(w) = EF_{\text{CLASS}}(w) - EF_{\text{GLEAM}} \tag{3}$$

where a unique soil moisture value ($w$) is considered for the entire root zone in CLASS. In order to assure convergence and reduce the number of iteration steps, two zero finding algorithms have been combined, namely the discrete midpoint method and the bisection method (Burden et al., 2016). We note that this procedure is analogous to the one used by Miralles et al. (2014) in which CLASS was steered to match the Bowen ratios (instead of the *EF*) derived from GLEAM. However for the results presented in Section 3, one-hourly values of *EF* from recently released reanalysis dataset ERA5 (Copernicus Climate Change Service (C3S), 2017) are used instead of daily values from GLEAM, which provided a higher temporal resolution and better performance statistics. Since gridded and consistent satellite datasets of cloud properties, advection, subsidence and radiation are not available for the long time span, we also make use of reanalysis data from ERA-Interim (Dee et al., 2011). Moreover, reanalysis data are also used to initialize the soil temperature in the morning. As an alternative to the specification of cloud cover, one could also directly specify the radiation inputs based on ERA-Interim or measurements from the Clouds and Earth's Radiant Energy System (CERES) onboard Terra and Aqua (Wielicki et al., 1996), which are available globally from the year 2001 onwards on a $1°$ regular grid. Unless specified differently, the data are used in a dynamic manner in the CLASS simulations based on the temporal resolution specified in the Table 1. Since the lateral forcing for the calculation of advection is only available at a coarse spatial ($0.75°$) and temporal (6-hourly) resolution, a footprint of $1°$ by $1°$ is taken for the ABL column model — centered over the sounding location — over which average parameters of the different datasets are calculated for the model input. Finally, the CLASS4GL framework implements a flexible interface to any NetCDF files based on the Python/Xarray software libraries that a user may be interested in adding, hence alternative datasets can easily be adopted.

| Data set | Variable | type | Horizontal resolution | Temporal resolution | Period | Reference |
|---|---|---|---|---|---|---|
| IGRA | Vertical profiles of potential temperature, wind and specific humidity | sounding | - | daily | 1900 – 2018 | Durre et al. (2006) |
| HUMPPA | *Idem* | sounding | - | 3-hourly | summer 2010 | Williams et al. (2011) |
| BLLAST | *Idem* | sounding | - | 3-hourly | summer 2011 | Pietersen et al. (2015) |
| GOAMAZON | *Idem* | sounding | - | 3-hourly | 2014 – 2015 | Martin et al. (2016) |
| GLEAM or ERA5 | Initial root-zone soil moisture and evaporative fraction | satellite / reanalysis | $0.25° \times 0.25°$ | daily | 1980 – 2017 | Martens et al. (2017); Miralles et al. (2011); Copernicus Climate Change Service (C3S) (2017) |
| IGBP-DIS | Soil hydrological properties (wilting point, water field capacity, saturated water content) | survey | $0.25° \times 0.25°$ | static | - | IGBP-DIS (2000) |
| GIMSS (AVHRR) | Leaf Area Index | satellite | $0.25° \times 0.25°$ | monthly | 1981 – 2015 | Liu et al. (2012) |
| HWSD | Clapp and Hornberger parameters, force–restore thermal parameters | survey | $0.085° \times 0.085°$ | static | - | Nachtergaele et al. (2009) |
| MOD44B (MODIS) | Fractional vegetation and albedo | satellite | 250m x 250m | yearly (here considered as static) | - | Hansen et al. (2005) |
| GLAS | Canopy height | satellite | $0.25° \times 0.25°$ | static | - | Simard et al. (2011) |
| ERA-Interim | Atmospheric forcing (cloud cover, advection, subsidence), initial soil temperature | reanalysis | $0.75° \times 0.75°$ | 6-hourly | 1979 – 2018 | Dee et al. (2011) |
| ERA-Interim or CERES | radiation components | reanalysis or satellite | $1° \times 1°$ | 6-hourly | 2001 – 2015 (CERES) | Wielicki et al. (1996) |

**Table 1.** Ancillary data sources of CLASS4GL.

| Vegetation | | | |
|---|---|---|---|
| *Symbol* | *Name* | *Unit* | *Default value or source* |
| LAI | Leaf area index of vegetated surface fraction | [-] | GIMSS (daily) |
| $r_{c,min}$ | Minimum resistance transpiration | [s m$^{-1}$] | 110 |
| $r_{s,soil,min}$ | Minimum resistance soil evaporation | [s m$^{-1}$] | 50 |
| $g_D$ | Vapour pressure deficit correction factor for surface resistance | [-] | 0 |
| $h_{\mathrm{can}}$ | Canopy height | [m] | GLAS (static) |
| $z_{0\mathrm{m}}$ | Roughness length for momentum | [m] | 0.1 x $h_{\mathrm{can}}$ (static) |
| $z_{0\mathrm{h}}$ | Roughness length for heat and moisture | [m] | 0.1 x $z_{0\mathrm{m}}$ (static) |
| $\alpha$ | Surface albedo | [-] | MOD44B (static) |
| $T_s$ | Initial surface temperature | [K] | ERA-Interim (three-hourly) |
| $T_{soil,1}, T_{soil,2}$ | Initial temperature of the top and deep soil layer | [K] | ERA-Interim (three-hourly) |
| $\omega_{\mathrm{sat}}$ | Saturated volumetric water content | [m$^3$ m$^{-3}$] | IGBP-DIS (static) |
| Soil | | | |
| *Symbol* | *Name* | *Unit* | *Default value or source* |
| $\omega_{\mathrm{fc}}$ | Volumetric water content field capacity | [m$^3$ m$^{-3}$] | IGBP-DIS (static) |
| $\omega_{\mathrm{wilt}}$ | Volumetric water content wilting point | [m$^3$ m$^{-3}$] | IGBP-DIS (static) |
| $EF$ | Evaporative fraction | [-] | ERA5 (hourly) or GLEAM (daily) |
| $\omega_{\mathrm{soil},1}, \omega_{\mathrm{soil},2}$ | Volumetric water content top and deep soil layer | [m$^3$ m$^{-3}$] | By iterative matching of EF |
| $c_{\mathrm{veg}}$ | Vegetation fraction | [-] | MOD44B (static) |
| $a, b, p$ | Clapp and Hornberger retention curve parameters | [-] | HWSD (static) |
| $C_{\mathrm{Gsat}}$ | Saturated soil conductivity for heat | [K m$^{-2}$ J$^{-1}$] | HWSD (static) |
| $C_{2,\mathrm{sat}}$ | Coefficient force term moisture | [-] | HWSD (static) |
| $C_{2,ref}$ | Coefficient restore term moisture | [-] | HWSD (static) |
| $\Lambda$ | Thermal diffusivity skin layer | [-] | 5.9 |
| Air | | | |
| *Symbol* | *Name* | *Unit* | *Default value or source* |
| $\beta$ | Ratio between buoyancy virtual heat and entrainment virtual heat | [-] | 0.2 |
| $\gamma_\theta$ | Initial lapse rate of potential temperature in the free atmosphere | [K m$^{-1}$] | From profile (IGRA) |
| $\gamma_q$ | Initial lapse rates of specific humidity in the free atmosphere | [kg kg$^{-1}$ m$^{-1}$] | *Idem* |
| $\gamma_u, \gamma_v$ | Initial lapse rates of zonal and meridional components in the free atmosphere | [m$^{-1}$] | *Idem* |
| $\Delta\theta_h$ | Initial jump for potential temperature between the mixed layer and free atmosphere | [K] | *Idem* |
| $\Delta\theta_q$ | Initial jump for specific humidity between the mixed layer and free atmosphere | [kg kg$^{-1}$] | *Idem* |
| $\Delta\theta_u, \Delta\theta_v$ | Initial jump for zonal and meridional wind components between the mixed layer and free atmosphere | [m s$^{-1}$] | *Idem* |
| $\langle \nabla_{\mathrm{hor}}\overline{\theta} \rangle$ | mixed-layer horizontal heat advection | [K s$^{-1}$] | ERA-Interim (6-hourly) |
| $\langle \nabla_{\mathrm{hor}}\overline{q} \rangle$ | mixed-layer moisture advection | [kg kg$^{-1}$ s$^{-1}$] | ERA-Interim (6-hourly) |
| $\langle \nabla_{\mathrm{hor}}\overline{u} \rangle$ | mixed-layer advection of zonal momentum | [m s$^{-2}$] | ERA-Interim (6-hourly) |
| $\langle \nabla_{\mathrm{hor}}\overline{v} \rangle$ | mixed-layer advection of meridional momentum | [m s$^{-2}$] | ERA-Interim (6-hourly) |
| $w_h$ | subsidence | [m s$^{-1}$] | ERA-Interim (6-hourly) |

**Table 2.** Main input parameters for CLASS4GL. The parameter specifications and source acronyms are explained in section 2.3 and table 1.

# 3  Results and discussion

The skill to replicate the evolution of the ABL as observed by the radiosondes is evaluated against the pre-processed and quality-controlled balloon soundings from (a) the three intensive research campaigns, and (b) the global operational IGRA dataset. The evaluation is done by comparing the modelled daytime tendencies of the mixed-layer height ($dh/dt$), potential
5  temperature ($d\theta/dt$) and specific humidity ($dq/dt$) against the corresponding tendencies observed from the balloon sounding pairs. Observed and modelled tendencies represent the mean diurnal change from the morning sounding to the afternoon sounding. It should be noted that the local time of the morning and afternoon soundings changes given that the launch times are often at 0 and 12 h UTC, and that the boundary-layer tendencies are not uniform over the course of the day. The resulting variety in the tendencies is taken into account in the simulations, since the model is initialized with the morning sounding while
10  the initial solar local time in the model is set equal to the sounding launch. The same happens for the end of the simulation at the time of the afternoon sounding. Therefore, the expected tendency for each launch or site (depending on the local time window being considered in the computation of that tendency) is equivalent for observations and model simulations, hence any biases related to launching times between the two are avoided. The campaign observations provide an *a priori* higher standard than the operational balloons in terms of accuracy and resolution during the balloon ascent and in terms of daytime sampling. Hence,
15  the evaluation against the 22 campaign soundings in our case serves as a control experiment of the model setup, initialization and forcing employed by CLASS4GL. In turn, validation of the model results against the 18k IGRA balloon sounding pairs serves as an overall evaluation of the suitability of CLASS4GL for the appraisal of the ABL behaviour observed and associated land–atmosphere feedbacks for different climate regimes.

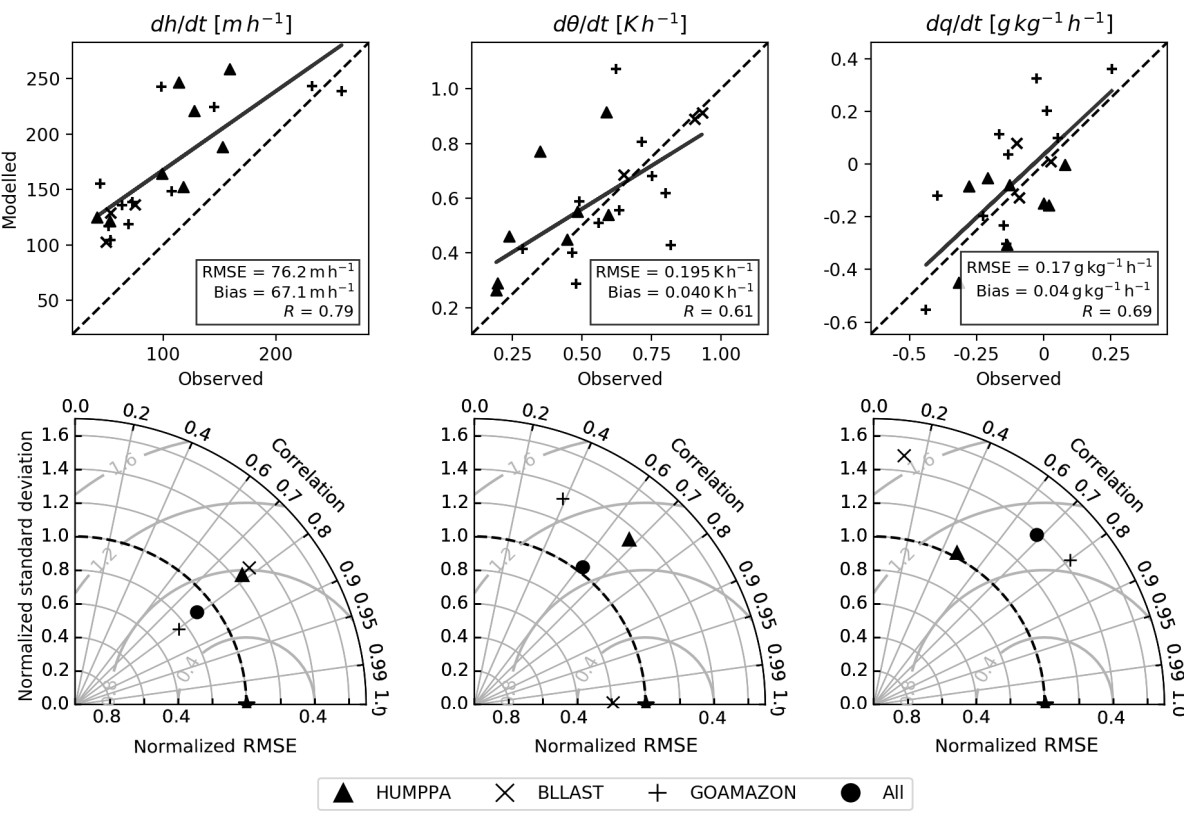

**Figure 3.** Performance statistics of diurnal changes in mixed-layer properties (mixed-layer height: $dh/dt$, potential temperature: $d\theta/dt$; specific humidity: $dq/dt$) during the three intensive observation campaigns: HUMPPA, BLLAST and GOAMAZON. The 1:1 line is shown as black dashed line.

The results of the model skill evaluation against the observations from the three intensive campaigns are summarized in Fig. 3. We find that the framework can reproduce the overall magnitude of the observed daytime tendencies ($dh/dt$, $d\theta/dt$ and $dq/dt$), with a bias of 67.1 m h$^{-1}$, 0.04 K h$^{-1}$ and 0.04 g kg$^{-1}$, respectively. The model further reproduces the overall differential response among the campaign days and sites, with a Pearson Correlation Coefficient ($R$) of 0.79, 0.61 and 0.69, for the three ABL tendencies, respectively. However, we could identify common model limitations over the three campaigns, particularly an overall (slight) positive bias in $dh/dt$ ($d\theta/dt$ and $dq/dt$), and an under(over)estimation of its (their) variability as indicated with a normalized standard deviation different from 1 in the Taylor plots (Fig. 3). The bias is expected to have multiple origins, including the ABL model and its physical concepts, the forcing data (convergence/advection, evaporation bias, cloud cover...), model tuning parameters (such as the entrainment ratio) and errors in the sounding observations used to initialize and validate the model. All these possible error sources should be investigated in further development of CLASS4GL. The results of the

three campaigns are useful as a first check of the model performance against high-quality observations. However, the sample size of 22 days over the three campaigns is too low to gain conclusive insights on whether these biases are of a systematic nature. In this respect, the validation of model performance against the 36k IGRA soundings from 121 different stations in different climate regions provides a more reliable and comprehensive assessment.

The simulated diurnal ABL tendencies show a similar accuracy when evaluated against the global IGRA sounding archive, despite the fact that these operational sounding have *a priori* lower quality standards than the campaign soundings; this applies to all three ABL tendencies shown in Fig. 4. Here, the magnitude of the bias is of the same order for the change in $\theta$ (-0.036 K h⁻¹) as for the intense observation campaigns, but is now negative. The biases are smaller for $h$ (10.1 m h⁻¹) compared to the campaigns and slightly larger for $q$ (0.06 g kg⁻¹). As for the research campaigns, the model is able to reproduce the variability

among the different operational sounding days, with Pearson correlation coefficients of 0.53, 0.82 and 0.54 for the diurnal tendencies of $h$, $\theta$ and $q$, respectively. In addition, the overall modelled range in $dh/dt$, $d\theta/dt$ and $dq/dt$ agrees well with the observed range, for which the departure of the modelled (normalized) standard deviation from the observed (normalized) standard deviation of each tendency is below 22%. This can be seen in the Taylor plots in Fig. 4 for which the centers of the open circles are between 0.78 and 1.22 of the normalized standard deviation. There is also a systematic underestimation

of the variability for $dh/dt$, $d\theta/dt$, but not for $dq/dt$. The negative bias in the temperature tendency and the positive bias in the humidity tendency lead to an overall net heat bias of 114 J kg⁻¹ h⁻¹. Similar as for the results in the campaigns, it is expected that such global biases have multiple origins, including biases in the net radiation (which is calculated by the model by prescribing the cloud cover), underestimation of ground heat storage to the soil, the entrainment rates, and/or the prescribed advection. Further research should investigate possible errors related to input datasets and validate them against independent

data (e.g., the available CERES data could be used to evaluate the net radiation). Performance statistics vary slightly with the climate region under scope, with e.g., the correlations for $d\theta/dt$ ranging between 0.70 and 0.89 depending on the climate regime. The highest model skill corresponds to warm temperate climates, while the lowest skill is found for humid regions, although the variability in model skill depends on the diagnosed variable and score metric. The similar model performance for the global IGRA archive (Fig. 4) compared to the intensive field campaigns (Fig. 3) gives us the confidence that the lower

quality standards of the operational measurements do not substantially hamper the framework performance, at least once the strict quality-based selection of radiosondes described in Section 2.2 is adopted.

     The results in Figs. 3 and 4 suggest that the overall model skill in reproducing the diurnal cycle of $/theta$ is higher than the skill to reproduce the mixed-layer growth and the cycle of $q$. Validation results also indicate an overall underestimation of the $\theta$ increase and overestimation of the change in $q$ as the day progresses. At the same time, the high mixed-layer growths are

underestimated. Additional sensitivity analyses aiming at lowering the $EF$ suggest that these deficiencies could result from an overall overestimation of surface evaporation, and an underestimation of surface sensible heat flux (not shown). This could relate in turn to errors in the $EF$ input dataset or in the vegetation parameters (vegetation cover and LAI) that the determine the partitioning between soil evaporation and transpiration. Moreover, the underrepresentation of extremes (e.g., very dry or very wet conditions) could relate to the coarse resolution of the surface soil parameters. Other factors, like the underestimation of

the entrainment may also contribute to the cold and wet bias of the model. Particularly, the modelled mixed-layer heat gain by

entrainment is positive (0.011 K m s$^{-1}$; i.e., warm air entrainment) and the average mixed-layer moisture gain by entrainment is negative (-2.24 x 10$^{-5}$ kg kg$^{-1}$ m s$^{-1}$; i.e., dry air entrainment) averaged over the whole sounding database. Intuitively, if entrainment was increased, due to (e.g.) a higher entrainment velocity, it would lead to further warming (in terms of $\theta$) and drying (in terms of $q$) of the mixed layer, hence it would act to reduce the overall bias.

5    Overall, validation results in Fig. 3 and 4 show that CLASS4GL provides pre-processing and modelling tools that can be used to represent the overall diurnal behaviour of the ABL. The model realistically represents the main characteristics of the global ABL diurnal evolution according to balloon soundings, including the observed variability around the globe. Several limitations are highlighted; when applying CLASS4GL, one should be aware of the fact that model parameters and assumptions, combined with input data uncertainty, can lead to the failure to simulate specific sounding profiles. The framework is in continuous

10   development, and it is expected that results will improve with higher resolution and accuracy of the forcing datasets, and with evolving model concepts of the ABL and land–atmosphere interface. The use of local parameterisations and higher quality forcing is encouraged for applications in specific regions.

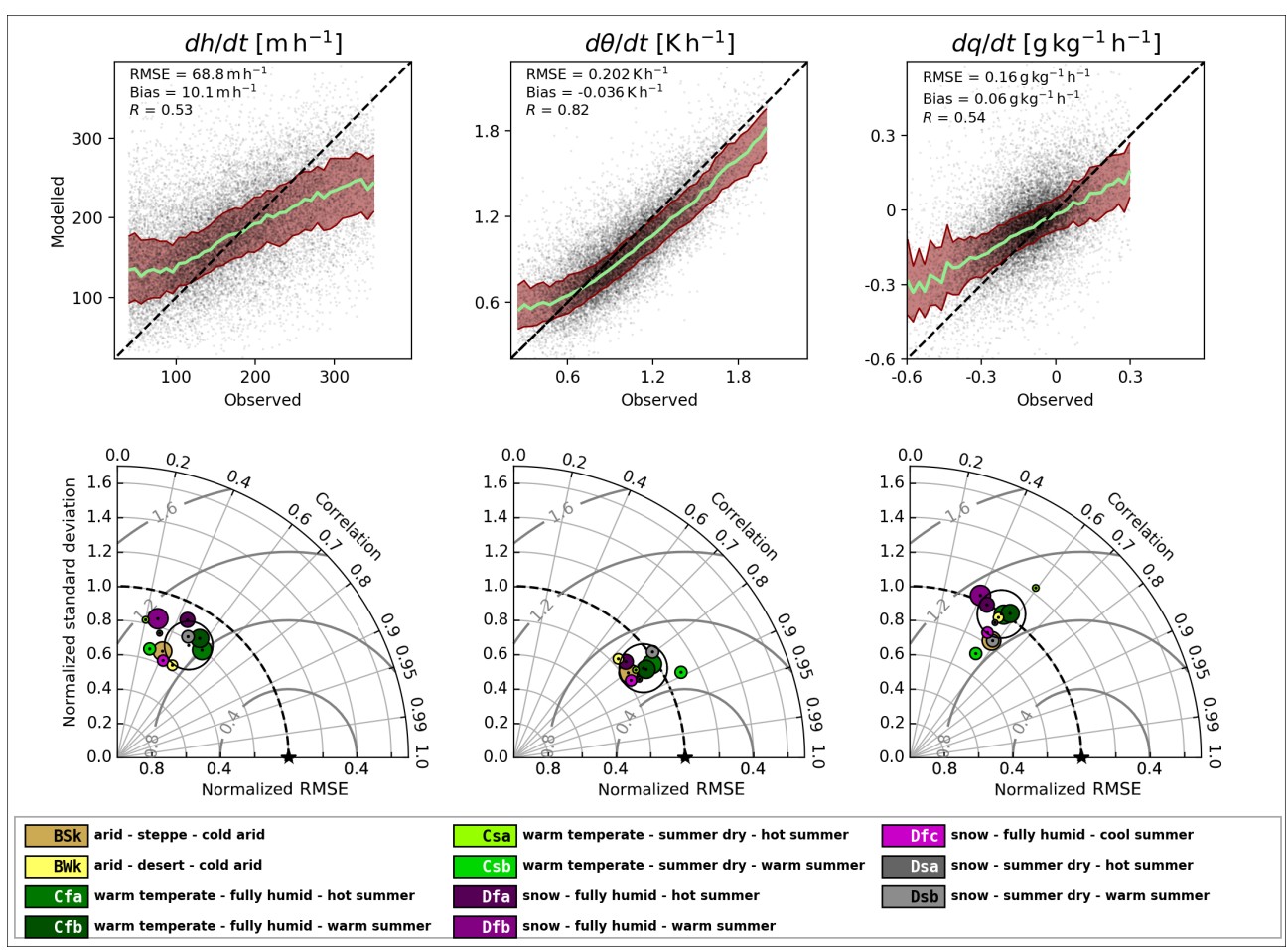

**Figure 4.** Model skill in reproducing diurnal changes in ABL properties. Shown are the tendencies of the mixed-layer height ($dh/dt$), potential temperature ($d\theta/dt$) and specific humidity ($dq/dt$), which are assessed by comparison of model simulations against the IGRA sounding data between 1981 and 2015. The upper plots show modelled versus observed data points (grey), and the corresponding median (green) and interquartile range (red) of the model. The 1:1 line is shown as black dashed line. The Taylor plots indicate the overall model performance as open circles, and the performance corresponding to each Köppen–Geiger climate class as colored bullets. Only climate regimes with more than > 200 soundings are illustrated. The size of the bullets is proportional to the amount of soundings for each particular climate class.

## 4 Conclusion and perspective

We have presented a novel interactive data platform, referred to as CLASS4GL (http://class4gl.eu), to automize the study of the diurnal ABL based on balloon radiosondes launched worldwide since the early '80s. The framework (a) mines balloon sounding data to initialize and validate the ABL bulk model CLASS, (b) links to a predefined but expandable set of global

datasets that are used to constrain the surface and large-scale atmospheric conditions in the CLASS simulations, (c) is successful in evaluating the skill to simulate the ABL globally, (d) provides a flexible and user-friendly interface for allowing extensive amount of experiments on supercomputer environments thanks to a low computational cost (e.g., a batch of 41, 000 diurnal simulations only takes $\sim$ 1 hour on a contemporary supercomputer infrastructure), and (e) strives for a community-driven architecture that allows to seamlessly share input datasets, experiments, analyses and model developments among the climate research community, and that also facilitates hackathons, workshops and educational activities. Validation results show an overal realistic representation of the diurnal evolution of mixed-layer height, potential temperature and specific humidity, using data from three observation campaigns and the IGRA operational sounding dataset.

The freely-available model framework offers new perspectives to foster the study of the diurnal evolution of the ABL and the associated land—atmosphere feedbacks:

– The fast software infrastructure allows any researcher to easily employ extensive global sensitivity experiments, in which both land and atmospheric parameters can be perturbed. That can be used to investigate the effect of land conditions and large-scale atmospheric forcing on the ABL evolution worldwide. Particularly, this inititive fits well within the context of LoCo activities (http://www.gewex.org/loco/) to, e.g., construct mixing diagrams and other metrics to understand land–atmospheric feedbacks. Moreover by integrating global information of precipitation, temperature and cloud statistics, one could further investigate whether particular combinations of surface and atmospheric conditions lead to ABL properties that favor or disfavor the occurrence of convection, clouds, precipitation, or extreme temperatures.

– Using the radio sounding simulations as reference, the framework can be employed to study the climate model representation of ABL dynamics and the associated land-–atmosphere feedbacks at the diurnal scales, and to evaluate satellite-based products and reanalysis data. Such a process-based validation can help improve climate models and assess the quality of satellite products intended to monitor the land–atmosphere interface.

– It could be used to challenge the added value of including novel mechanistic concepts, such as the dynamic representation of soil and vegetation interacting with the carbon cycle ($CO_2$ exchange, carbon stock, air $CO_2$ levels, etc.), atmospheric chemistry (VOCs, aerosols, ozone, etc.), vegetation dynamics and water stress (Combe et al., 2016), or urbanization (Droste et al., 2018; Wouters et al., 2016).

– Finally, the ABL evolution and the associated land–atmosphere interactions could be extracted from climate projections and land-cover climate scenarios. This way, one may determine the local drivers of shifting (extreme) weather under climate change, hence provide a better process-understanding. When integrating future land use scenarios, it may foster the development of more effective climate adaptation strategies, e.g., by quantifying the mitigatory potential of land use change that could alleviate the escalation of mixed-layer temperatures during heatwaves.

**Code availability**

CLASS4GL freely available as a Python module and is conveyed under GNU General Public license version 3 (GPLv3). The general information, code and tutorials for running the software is maintained at http://class4gl.eu. The presented model version for CLASS4GL is 1.0. All comments, questions, suggestions and critiques regarding the functioning of the Python routine can
be directed to the author of this paper.

**Author contribution**

H.W., D.G.M. and I.Y.P. proposed the initial concept and data flow. H.W. developed the code and did the analyses. C.C.v.H., J.V.-G.d.A., A.J.T. and J.A.S. provided conceptual support regarding land–atmosphere interactions and ABL physics. H.W. and D.G.M. led the writing with contributions from all coauthors. V.M. and I.Y.P. proposed improvements to the user interface.
All authors contributed to the design of the experiments and the interpretation of the results.

*Acknowledgements.* This research was funded by the European Research Council (ERC) under Grant Agreement No. 715254 (DRY–2–DRY). BLLAST field experiment was made possible thanks to the contributions of several institutions and supports: INSU-CNRS (Institute National des Sciences de l'Univers, Centre National de la Recherche Scientifique, LEFE-IDAO program), Météo France, Observatoire Midi-Pyrénées (University of Toulouse), EUFAR (EUropean Facility for Airborne Research) and COST ES0802 (European Cooperation in the
field of Science and Technology). The field experiment would not have occurred without the contribution of all participating European and American research groups, which all have contributed in a significant amount (see http://bllast.sedoo.fr/supports/). BLLAST field experiment was hosted by the instrumental site of Centre des Recherches Atmosphérique, Lannemezan, France (Observatoire Midi-Pyrénées, Laboratoire d'Aérologie). BLLAST data are managed by SEDOO, from Observatoire Midi-Pirénées. We would like to thank J. Williams et al. and S.T. Martin et al. for providing the campaign data for HUMPPA and GOAMAZON, respectively. We thank Florian Cochard and Pierre Gentine
for the radiosonde and profile analysis. We also thank Dominik Schumacher and Joke De Meester for employing independent tests of the software code and tutorials. The computational resources and services used in this work were provided by the VSC (Flemish Supercomputer Center), funded by the Research Foundation - Flanders (FWO) and the Flemish Government – department EWI.

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
