# Peer review of "Atmospheric boundary layer dynamics from balloon soundings worldwide: CLASS4GL v1.0"

_Geoscientific Model Development, 2019_

## Referee Comment (RC1) · Anonymous Referee #1 · 4 Apr 2019

General Comments

This is a well-written manuscript which documents a powerful new software tool which the authors are making publicly available. This tool should allow researchers to perform extensive experiments related to boundary layer growth and development, including sensitivity to land surface and atmospheric inputs and parameters. The input datasets are global and extend back to 1981, allowing for easy application of experiments across climate regimes and seasons, and allowing users to test the representation of boundary layer dynamics in climate and earth system models. The authors include analysis of an initial experiment to demonstrate the first-order performance of the model. This

projects looks like it could be extremely beneficial to the broad scientific community. I am fully in favor of this manuscript being published in Geoscientific Model Development. I document a few very minor suggestions below.

Minor Suggestions

1. Section 2.1, line 15: "for which one also adds the entrainment flux driven by shear": I'm not quite sure what this phrase means. Do you mean that the component driven by shear is included in the 0.2*(buoyancy flux) term, or that it should be added to this term? If it's the latter, you should change the word "for" to "to".

2. Line 18: Which parameters in the Penman-Monteith and other empirical equations are fixed and which are locally and/or seasonally determined from the input datasets?

3. Figure 2: It took me a while to find the big X for the BLLAST experiment location, in part because much of the X is on top of country lines. Perhaps you can use a different symbol.

4. Section 3, line 3: When you first mention daytime tendencies, can you clarify what time period the resultant values are averaged over? I imagine it is from sunrise through the time of the second sounding. Since the local timing of this second sounding is at different times of day in different longitudes, might this introduce a spatial bias since BL growth rates are not uniform over the course of the day?

5. Page 12, line 2: Here you mention the observed daytime tendencies when you are discussing the results from the three intensive campaigns. Are the results you show actually subdaily averages since you have more than two soundings per day?

6. Figure 3: the correlation plots are quite busy, making it a touch hard to find the three symbols of interest. Maybe you can make the grey lines a little bit lighter grey so the symbols are easier to see.

7. Page 13, line 8: It is not clear to me where the 22% value comes from. Please clarify.

---

## Referee Comment (RC2) · Anonymous Referee #2 · 5 Apr 2019

The paper presents a significant advancement in producing both useful accessible boundary layer data from radiosondes, and a nice marriage with a simple ABL model to produce continuous ABL data constrained by analyses, with open-source software.

I was not able to fully run the CLASStGL software myself. On a Mac using MacPorts, the PyYAML was not available and I downloaded directly from the website - there were issues recognizing the CLoader option - apparently a version inconsistency. But I will follow through as I would like to use this tool.

Regarding the manuscript, I suggest only minor changes are needed (editorial and regarding content), as outlined below:

[Figure]

P3 L14: Use "automates" instead of "automises". Likewise on P4 L7.

P4 L3: Change "dirunal" to "diurnal"

P5, L10-11: It is a common assumption that the heat, moisture and momentum content of the ABL are perfectly mixed, but of course there will be mean vertical gradients, especially near the entrainment zone and the surface. In other words, the gradients here are a little weaker than for a well-mixed ABL, which may be compensated by other parameter choices. What would be the effect of specifying more realistic but still simple tails (e.g., exponential or even linear) of theta, q and V at the top and bottom of the ABL? This will relate to comments below regarding apparent biases.

P7 L4-10: Please state how many (or what percentage) of the 42,000 profiles are excluded for each reason (lacking both 00 and 12UTC soundings vs. non well-mixed profiles? The first seems a hard criterion, but exactly how well-mixed is that criterion and what if it is relaxed?

P7 L12: Change "says" to "days".

P7 L13-17: Are there clear discrepancies between the behavior and/or statistics of gap-filled (model) versus observationally driven results? I assume you have looked at this - a caveat might be warranted here.

P10 L8: Change "reassure" to "assure".

Figure 3: There is only a circle (All) for dq/dt - not the other rates. Is something missing?

P12 L6: Here I start wondering about the sources of biases and if you have been able to examine them. For dq/dt, a positive evaporation bias, excessive low-level moisture flux convergence (in the boundary conditions) or too little entrainment of dry air could each explain this. Has it been investigated? Is it likely a problem with the model or forcing data?

Fig 4 and associated text: If the heating and moistening rates are converted to J/kg/h by

multiplying by C_p and lambda_v respectively, we get that the heating bias is -52 J/kg/h but the positive moistening bias is 175 J/kg/h. The discrepancy is 123 J/kg/h – again there could be multiple sources of this. First thought is net radiation, but excessive ground heat flux from the soil, advection (convergence) or entrainment could all be reasons. Any idea about the source of this net energy bias?

P13 L25-29: Related to above, a nice speculation on causes, but atmospheric models including reanalyses tend to have too much surface net radiation due to cloud errors and lack or proper representation of aerosol effects. R_Net or the input ERA-I radiation should be validated against independent data (e.g., the available CERES data) as a sanity check.

―――――――――――――――――――

---

## Author Comment (AC1) · 23 Apr 2019

*General Comments*

*This is a well-written manuscript which documents a powerful new software tool which the authors are making publicly available. This tool should allow researchers to perform extensive experiments related to boundary layer growth and development, including sensitivity to land surface and atmospheric inputs and parameters. The input datasets are global and extend back to 1981, allowing for easy application of experiments across climate regimes and seasons, and allowing users to test the representation of boundary layer dynamics in climate and earth system models. The authors include analysis of an initial experiment to demonstrate the first-order performance of the model. This projects looks like it could be extremely beneficial to the broad scientific community. I am fully in favor of this manuscript being published in Geoscientific Model Development. I document a few very minor suggestions below.*

We would like to thank the referee for providing their review of the manuscript, and we are very glad to note the appreciation of the software's potential. We also acknowledge their comments, especially for improving the clarity of used methodologies, input data and results,  and we provide a point-by-point answer below. The changes to the manuscript are provided as quoted text, which will be included in the next revised version of the manuscript.

*Minor Suggestions*

*1. Section 2.1, line 15: "for which one also adds the entrainment flux driven by shear": I'm not quite sure what this phrase means. Do you mean that the component driven by shear is included in the 0.2*(buoyancy flux) term, or that it should be added to this term? If it's the latter, you should change the word "for" to "to".*

It is the latter, namely the component driven by shear is added to the buoyancy flux afterwards. Hence, we change "for" to "to". In the revised manuscript, it will read as: "Entrainment flux is calculated as a fixed fraction (0.2) of the buoyancy flux, **to** which one also adds the entrainment flux driven by shear."

*2. Line 18: Which parameters in the Penman-Monteith and other empirical equations are fixed and which are locally and/or seasonally determined from the input datasets?'*

Here is a table for the most important parameters in the Penman-Monteith and other empirical equations regarding to vegetation, soil and the air. It will be included as a second table in the revised manuscript:

**Vegetation**

| Symbol | Name | Unit | Default value or source |
|---|---|---|---|
| LAI | Leaf area index of vegetated surface fraction | [-] | GIMSS |
| $r_{c,min}$ | Minimum resistance transpiration | [s m$^{-1}$] | 110 |
| $r_{s,soil,min}$ | Minimum resistance soil evaporation | [s m$^{-1}$] | 50 |
| $g_D$ | Vapour pressure deficit correction factor for surface resistance | [-] | 0 |
| $h_{can}$ | Canopy height | [m] | GLAS |
| $z_{0m}$ | Roughness length for momentum | [m] | 0.1 x $h_{can}$ |
| $z_{0h}$ | Roughness length for heat and moisture | [m] | 0.1 x $z_{0m}$ |
| α | Surface albedo | [-] | MOD44B |
| $T_s$ | Initial surface temperature | [K] | ERA-Interim |
| $T_{soil.1}$ | Temperature top soil layer | [K] | ERA-Interim |
| $T_{soil.2}$ | Temperature deeper soil layer | [K] | ERA-Interim |
| $\omega_{sat}$ | Saturated volumetric water content | [m$^3$ m$^{-3}$] | IGBP-DIS |

**Soil**

| Symbol | Name | Unit | Default value or source |
|---|---|---|---|
| $\omega_{fc}$ | Volumetric water content field capacity | [m$^3$ m$^{-3}$] | IGBP-DIS |
| $\omega_{wilt}$ | Volumetric water content wilting point | [m$^3$ m$^{-3}$] | IGBP-DIS |
| EF | Evaporative fraction | [-] | ERA5 |
| $\omega_{soil,1}$ | Volumetric water content top soil layer | [m$^3$ m$^{-3}$] | By iterative matching of EF |
| $\omega_{soil,2}$ | Volumetric water content deeper soil layer idem | | *Idem* |
| $c_{veg}$ | Vegetation fraction | [-] | MOD44B |
| a | Clapp and Hornberger retention curve parameter | [-] | HWSD |
| b | Clapp and Hornberger retention curve parameter | [-] | HWSD |
| p | Clapp and Hornberger retention curve parameter | [-] | HWSD |
| $C_{Gsat}$ | Saturated soil conductivity for heat | [K m$^{-2}$ J$^{-1}$] | HWSD |
| $C_{2,sat}$ | Coefficient force term moisture | [-] | HWSD |
| $C_{2,ref}$ | Coefficient restore term moisture | [-] | HWSD |
| Λ | Thermal diffusivity skin layer | [-] | 5.9 |

**Air**

| Symbol | Name | Unit | Default value or source |
|---|---|---|---|
| β | Ratio between buoyancy virtual heat and entrainment virtual heat | [-] | 0.2 |

| | | | |
|---|---|---|---|
| $\gamma_\theta$ | Initial lapse rate of potential temperature in the free atmosphere | [K m$^{-1}$] | From profile (IGRA) |
| $\gamma_q$ | Initial lapse rate of specific humidity in the free atmosphere | [kg kg$^{-1}$ m$^{-1}$] | *Idem* |
| $\gamma_u$ | Initial lapse rate of zonal wind in the free atmosphere | [s$^{-1}$] | *Idem* |
| $\gamma_v$ | Initial lapse rate of meridional wind in the free atmosphere | [s$^{-1}$] | *Idem* |
| $\Delta\theta_h$ | Initial temperature jump between the mixed layer and free atmosphere | [K] | *Idem* |
| $\Delta\theta_q$ | Initial specific humidity jump between the mixed layer and free atmosphere | [kg kg$^{-1}$] | *Idem* |
| $\Delta\theta_u$ | Initial specific zonal wind jump between the mixed layer and free atmosphere | [m s$^{-1}$] | *Idem* |
| $\Delta\theta_v$ | Initial specific meridional wind jump between the mixed layer and free atmosphere | [m s$^{-1}$] | *Idem* |

Table2. Surface input parameters for CLASS4GL. The parameter specifications and source acronyms are explained in section 2.3, see also table 1.

*3. Figure 2: It took me a while to find the big X for the BLLAST experiment location, in part because much of the X is on top of country lines. Perhaps you can use a different Symbol.*

The symbol for the BLLAST location is now replaced with a thick cross. It will now appear as follows, which should make it more traceable:

[Figure]

*4. Section 3, line 3: When you first mention daytime tendencies, can you clarify what time period the resultant values are averaged over? I imagine it is from sunrise through the time of the second sounding.*

The tendencies are averaged from the morning sounding to the afternoon sounding. This will appear explicitly in the revised text as follows:

"The evaluation is done by comparing the modelled daytime tendencies of the mixed-layer height (dh/dt), potential temperature (dθ/dt) and specific humidity (dq/dt) against the corresponding tendencies observed from the balloon sounding pairs. **Observed and modelled tendencies represent the mean diurnal change from the morning sounding to the afternoon sounding.**"

*Since the local timing of this second sounding is at different times of day in different longitudes, might this introduce a spatial bias since BL growth rates are not uniform over the course of the day?*

Ideally, one would always have a sounding at the same local time in the morning and in the afternoon. Since the launching times are based on UTC, this is obviously not the case. So we agree that the common local time of the sounding launch depends on longitude, and also that the ABL growth is certainly not uniform over the course of the day. However, the latter is taken into account, since the model is always initialized with the morning sounding for which the initial local model time is set equal to the sounding launch, and the same is true for the afternoon sounding. So it can be concluded that the expected tendency for each launch or site (depending

on the local time window being considered in the computation of that tendency) is equivalent for observations and models, hence any biases related to launching times between the two are avoided.

This will be clarified with the following additional text (it will be located just after the previous text):

"It should be noted that the local time of the morning and afternoon soundings changes given that the launch times are often at 12 and 0 UTC, and that the boundary-layer tendencies are not uniform over the course of the day. The resulting variety in the tendencies is taken into account in the simulations, since the model is initialized with the morning sounding while the initial solar local time in the model is set equal to the sounding launch. The same happens for the end of the simulation at the time of the afternoon sounding. Hence, the expected tendency for each launch or site (depending on the local time window being considered in the computation of that tendency) is equivalent for observations and models, hence any biases related to launching times between the two are avoided"

*5. Page 12, line 2: Here you mention the observed daytime tendencies when you are discussing the results from the three intensive campaigns. Are the results you show actually subdaily averages since you have more than two soundings per day?*

Yes, as clarified above, the results reflect the diurnal tendencies averaged over the time span between the morning and afternoon sounding, hence depend on the sounding launch times. In case there are multiple soundings retained during a particular day which is especially the case for the campaigns, the sounding closest to sunrise is taken for the initialization in the morning, and the latest sounding for the validation in the afternoon. This was not mentioned explicitly in the text. This clarification will now be added in the methodology section '2.2. Automated balloon data mining':

"... Finally, the afternoon radiosonde profile on the same day needs to occur between local noon and 1 h before sunset **(defined as the time when the incoming shortwave radiation at the top of the atmosphere becomes zero)**, and at least 4 h after the model initialization in the morning **so that a sufficiently large model time span is considered**. **In case there are more than two soundings retained during a particular day which especially occurs during the campaigns, the sounding closest to sunrise is taken for the initialization in the morning, and the latest sounding for the validation in the afternoon**. ..."

*6. Figure 3: the correlation plots are quite busy, making it a touch hard to find the three symbols of interest. Maybe you can make the grey lines a little bit lighter grey so the symbols are easier to see.*

The grid lines are now lighter which increased the visibility:

[Figure]

The updated figure will be included in the revised manuscript.

*7. Page 13, line 8: It is not clear to me where the 22% value comes from. Please clarify.*

The 22% value refers to the departure of the (normalized) standard deviation of the model output from the (normalized) standard deviation of the observations. This can be seen on the Taylor plots in Fig. 4 (figure pasted below) for which the centers of the open circles are between 0.78 and 1.22 of the normalized standard deviation. This will be made clear in the text as follows:

"In addition, the overall modelled range in dh/dt, dθ/dt and dq/dt agrees well with the observed range, with departures from the standard deviation of the observations below 22% – see Taylor plots in Fig. 4., **for which the departure of the (normalized) standard deviation of the respective modelled parameters results from the (normalized) standard deviation of the observed parameters is below 22%. This can be seen on the Taylor plots in Fig. 4 for the centers of the open circles are between 0.78 and 1.22 of the normalized standard deviation.**"

[Figure]

---

## Author Comment (AC2) · 23 Apr 2019

*The paper presents a significant advancement in producing both useful accessible boundary layer data from radiosondes, and a nice marriage with a simple ABL model to produce continuous ABL data constrained by analyses, with open-source software.*

We would like to thank the referee for providing their review of the manuscript, and we are very glad regarding the appreciation of the software's potential. We also appreciate the comments, especially the suggestions related to possible sources of model biases. We provide a point-by-point answer below. The changes to the manuscript are provided as quoted text, which will be included in the next revised version of the manuscript.

*I was not able to fully run the CLASS4GL software myself. On a Mac using MacPorts, the PyYAML was not available and I downloaded directly from the website - there were issues recognizing the CLoader option - apparently a version inconsistency. But I will follow through as I would like to use this tool.*

Thank you for testing software! The goal is to have a platform-independent software, so we strive to make it work on all platforms including Mac systems. As a solution on Mac, we would like to suggest to try either a Python environment with anaconda (as explained on https://class4gl.eu/?page_id=105) or Pycharm+homebrew. In case of pycharm+homebrew, these are the needed steps to install the CLoader module:

```
brew install libyaml-dev
pip install pyyaml
```

Please note that 'brew install libyaml' (so without '-dev') will not work. A similar solution may exist in case of your current Python environment using MacPorts. The CLoader is required to read yaml files 10–100 times faster, but depends on modules written in the C language.

*Regarding the manuscript, I suggest only minor changes are needed (editorial and regarding content), as outlined below:*

*P3 L14: Use "automates" instead of "automises". Likewise on P4 L7.*
*P4 L3: Change "dirunal" to "diurnal"*

Thanks for identifying the typos. They will be corrected in the revised manuscript.

*P5, L10-11: It is a common assumption that the heat, moisture and momentum content of the ABL are perfectly mixed, but of course there will be mean vertical gradients, especially near the entrainment zone and the surface. In other words, the gradients here are a little weaker than for a well-mixed ABL, which may be compensated by other parameter choices. What would be the effect of specifying more realistic but still simple tails (e.g., exponential or even linear) of theta, q*

*and V at the top and bottom of the ABL? This will relate to comments below regarding apparent biases.*

It is true that the ABL model considers a perfectly-mixed ABL with values of potential temperature, specific humidity and wind speed that are constant throughout the ABL, whereas the entrainment zone is represented as a jump between the ABL values of and the free atmosphere values, and the surface layer as a analytic profile between ABL values and the surface values. Other gradients within of the ABL are not explicitly represented. The Monin-Obukhov similarity theory is employed for calculating analytic surface layer profiles and the gradient transport in surface layer in an implicit way as a replacement for a more explicit representation. For the entrainment zone, the heat entrainment ratio ($\beta$) of 0.2 (the ratio of heat entrainment to heating through the surface layer) is considered and the additional entrainment by wind shear, based on observations and large eddy simulations (Vilà-Guerau de Arellano et al., 2015). More realistic tails at the top and bottom of the ABL are not yet considered in the ABL model, hence, it is not possible to quantify their effect and the possible associated biases in the model. Therefore, one would require a dedicated study for which one needs substantial changes to the ABL model formulations. We are aware about the model limitations and associated uncertainties, and about the need for more research employing more realistic profiles. This will be mentioned more explicitly in the revised manuscript as follows:

"The use of the mixed-layer equations implies that the turbulence inside the ABL is not explicitly solved, and assumes that the potential temperature ($\theta$), specific humidity (q) and wind components are homogeneous within the ABL. This assumption tends to be supported by the efficient turbulent mixing under convective conditions (Bauer, 1908). At the top of the ABL, the entrainment of heat and moisture is parameterized by a jump of $\theta$, q and wind components over an infinitesimally small height, which are initialized with a constant lapse rate with height in the overlying free atmosphere. Entrainment flux is calculated as a fixed fraction (0.2) of the buoyancy flux, for which one also adds the entrainment flux driven by shear. An important feature of the model is the possibility to represent the subsidence coupled to the entrainment process at the inversion zone (Vilà-Guerau de Arellano et al., 2015). The surface–atmosphere turbulent exchanges for momentum, heat, and moisture **in the surface layer** are calculated considering their aerodynamic resistances. These are calculated in an iterative way assuming constant values for aerodynamic roughness lengths, while applying correction factors for non-neutral stratification of the atmospheric surface-layer (Paulson, 1970) according to the Monin–Obukhov similarity theory (Monin and Obukhov, 1954). **It should be kept in mind that more realistic profiles with explicit ABL gradients for temperature, humidity and wind speed – especially at the top (entrainment zone) and bottom (surface layer) of the ABL – are not yet considered by the model. In order to tackle these limitations and associated uncertainties, more research is needed employing more realistic profiles.**"

P7 L4-10: Please state how many (or what percentage) of the 42,000 profiles are excluded for each reason (lacking both 00 and 12UTC soundings vs. non well-mixed profiles? The first seems a hard criterion, but exactly how well-mixed is that criterion and what if it is relaxed?

The criterion for a well-mixed profile is that the root mean square error of the profile measurements in the boundary layer is lower than 1.5°C. This information will be added to the manuscript. In addition, we will provide the statistics on the reasons of profile retainment for each filtering step. Therefore, paragraph 2.2 will be revised and it will read in the revised manuscript as follows (additional information is indicated in bold):

"2.2 Automated balloon data mining

[revised manuscript text omitted]

*P7 L12: Change "says" to "days".*

*Done*

*P7 L13-17: Are there clear discrepancies between the behavior and/or statistics of gap-filled (model) versus observationally driven results? I assume you have looked at this - a caveat might be warranted here.*

At this stage, we have only implemented the possibility of extracting profiles from continuous/gap-filled  datasets (reanalysis, satellite-based products, and Earth system models...). The discrepancies between *gap-filled versus observationally driven ABL model results have not been done yet. We agree that this is an important caveat, so the user should be warned that such a validation is needed as soon as a gap-filled dataset is employed. Hence, the following text will be added to the revised manuscript:*

"This alternative to the use of sounding data holds great promise for spatially-explicit climatological ABL studies and multi-annual trend assessments. **It should be noted that ABL model simulations using continuous/gap-filled datasets may deviate from those using**

the observations. **Hence, an additional validation is needed as soon as such datasets are employed, in which one should compare the gap-filled datasets and the observations and the error propagation on the ABL model simulations**. Such an in-depth evaluation against the available sounding pairs can be done using the  presented framework ."

Done.

*Figure 3: There is only a circle (All) for dq/dt - not the other rates. Is something missing?*

They were indeed missing. All symbols will appear in the revised manuscript, as shown in the figure below:

[Figure]

*P12 L6: Here I start wondering about the sources of biases and if you have been able to examine them. For dq/dt, a positive evaporation bias, excessive low-level moisture flux convergence (in the boundary conditions) or too little entrainment of dry air could each explain this. Has it been investigated? Is it likely a problem with the model or forcing data?*

We also expect that the bias have multiple origins, so both the ABL model (physical concepts) and its forcing data (convergence/advection, evaporation bias, cloud cover...) but also model tuning parameters (eg., entrainment ratio) and errors in the sounding observations used to initialize and validate the model, and all of these possible errors should be investigated in the further development of CLASS4GL. We will make this clear by adding the following text in the revised manuscript as follows:

"**The bias is expected to come from multiple origins, including the ABL model and its physical concepts, the forcing data (convergence/advection, evaporation bias, cloud cover…, see Table 1), model tuning parameters (such as the entrainment ratio, see Table 2) and errors in the sounding observations used to initialize and validate the model. All these possible error sources should be investigated in further development of CLASS4GL**"

The sources of biases from forcing data is supported by improved model statistics when using 1-hourly values (ERA5) instead of daily values (GLEAM) for evaporative fraction for the campaigns (see figure above) and the global results (see figure below, which is discussed further below). Particularly, the bias of dq/dt for the campaigns is now 0.04 g kg$^{-1}$ h$^{-1}$ (see figure above), whereas it was much higher in the previous results 0.17 g kg$^{-1}$ h$^{-1}$. The new results will also be included in the revised manuscript. Using the updated results as shown with the figures above, we have modified the text accordingly, eg.,:

"However, we could identify common model limitations over the three campaigns, see Fig. 3. . **This includes an overall (slight) positive bias in dh/dt (dθ/dt and dq/dt), and a under(over)estimation of its (their) variability** as indicated with **a** normalized standard deviation **different from**  1 in the Taylor plots (Fig. 3)**."**

*Fig 4 and associated text: If the heating and moistening rates are converted to J/kg/h by multiplying by C_p and lambda_v respectively, we get that the heating bias is -52 J/kg/h but the positive moistening bias is 175 J/kg/h.*

Using the 1-hourly ERA5 data as forcing for evaporative fraction as discussed above, we now get an overall better model performance for the IGRA global soundings (see figure below): particularly, the global negative bias in dβ/dt has been reduced from -0.052K/h (-52 J kg$^{-1}$ h$^{-1}$) to -0.036K h$^{-1}$ (-36 J kg$^{-1}$ h$^{-1}$), and the positive bias in dq/dt has been reduced from 0.07 g kg$^{-1}$ h$^{-1}$ (or 175 J kg$^{-1}$ h$^{-1}$) to 0.06 g kg$^{-1}$ h$^{-1}$ (or 150 J kg$^{-1}$ h$^{-1}$). Pearson correlation coefficients have slightly improved (0.54/0.79/0.52 -> 0.53/0.82/0.54), and all RMSEs are slightly lower as well (72.3 m h$^{-1}$ / 0.211 K h$^{-1}$ / 0.17 g kg$^{-1}$ h$^{-1}$ -> 68.8m h$^{-1}$ / 0.202 K h$^{-1}$ / 0.16 g kg$^{-1}$ h$^{-1}$).

[Figure]

These results will be updated in the revised manuscript, and the text and discussions will be modified accordingly.

*The discrepancy is 123 J/kg/h – ...*

With the updated results mentioned above, the discrepancy has now been slightly reduced from 123 J kg$^{-1}$ h$^{-1}$ to 114 J kg$^{-1}$ h$^{-1}$.

*… again there could be multiple sources of this. First thought is net radiation, but excessive ground heat flux from the soil, advection (convergence) or entrainment could all be reasons. Any idea about the source of this net energy bias?*

*P13 L25-29: Related to above, a nice speculation on causes, but atmospheric models including reanalyses tend to have too much surface net radiation due to cloud errors and lack or proper representation of aerosol effects. R_Net or the input ERA-I radiation should be validated against independent data (e.g., the available CERES data) as a sanity check.*

In line with the discussion above, we expect that the source of the net energy bias have multiple origins, including the ABL model and its physical concepts, the forcing data, model tuning parameters and errors in the sounding observations. We agree with the suggestions of the

referee that this could be especially due to biases in the net radiation (which is calculated by the model by prescribing the cloud cover), the ground heat flux, the entrainment rates, and/or the prescribed advection. All this will be discussed in the revised manuscript as follows:

"In addition, the overall modelled range in dh/dt, dθ/dt and dq/dt agrees well with the observed range, with departures from the standard deviation of the observations below 22% – see Taylor plots in Fig. 4. There is also a systematic underestimation of the variability for dh/dt, dθ/dt, but not for dq/dt. **The negative bias in the temperature tendency and the positive bias in the humidity tendency leads to an overall net heat bias of 114 J kg$^{-1}$ h$^{-1}$. Similar as for the results in the campaigns, it is expected that such global biases have multiple origins, including biases in the net radiation (which is calculated by the model by prescribing the cloud cover), underestimation of ground heat storage to the soil, the entrainment rates, and/or the prescribed advection*. Further research should investigate possible errors related to input datasets and validate them against independent data (e.g., the available CERES data could be used to evaluate the net radiation).*"

---

## Author Response (AR1)

et al.*
*Anonymous Referee #1*

*General Comments*
*This is a well-written manuscript which documents a powerful new software tool which the
authors are making publicly available. This tool should allow researchers to perform extensive
experiments related to boundary layer growth and development, including sensitivity to land
surface and atmospheric inputs and parameters. The input datasets are global and extend back
to 1981, allowing for easy application of experiments across climate regimes and seasons, and
allowing users to test the representation of boundary layer dynamics in climate and earth
system models. The authors include analysis of an initial experiment to demonstrate the
first-order performance of the model. This projects looks like it could be extremely beneficial to
the broad scientific community. I am fully in favor of this manuscript being published in
Geoscientific Model Development. I document a few very minor suggestions below.*

We would like to thank the referee for providing his/her review of the manuscript, and we are
very glad to note the appreciation of the software's potential. We also acknowledge the
comments, especially for improving the clarity of used methodologies, input data and results,
and we provide a point-by-point answer below. Text of the revised manuscript is "quoted" in
which the changes are provided in **bold** and removed text as . The location of
text changes in the manuscript are provided in red. Unless specified otherwise, page and line
numbers refer to the revised manuscript version with track changes.

*Minor Suggestions*
*1. Section 2.1, line 15: "for which one also adds the entrainment flux driven by shear": I'm not
quite sure what this phrase means. Do you mean that the component driven by shear is
included in the 0.2\*(buoyancy flux) term, or that it should be added to this term? If it's the latter,
you should change the word "for" to "to".*

It is the latter, namely the component driven by shear is added to the buoyancy flux afterwards.
Hence, we change "for" to "to". It now reads as: "Entrainment flux is calculated as a fixed
fraction (0.2) of the buoyancy flux, **to** which one also adds the entrainment flux driven by shear."
P5R16

*2. Line 18: Which parameters in the Penman-Monteith and other empirical equations are fixed
and which are locally and/or seasonally determined from the input datasets?'*

In order to answer this question, we now provide a table for the most important parameters in
the Penman-Monteith and other empirical equations regarding to vegetation, soil and the air in
the manuscript. The table indicates either whether it's a fixed parameter or specified from an
external data source. In case of the latter, it is also indicated whether it's a static or
time-dependent data. It is now included as a second table in the revised manuscript (P12 of
revised manuscript without track changes), see also below:

"

**Vegetation**

| Symbol | Name | Unit | Default value or source |
|---|---|---|---|
| LAI | Leaf area index of vegetated surface fraction | [-] | GIMSS (daily) |
| $r_{c,min}$ | Minimum resistance transpiration | [s m$^{-1}$] | 110 |
| $r_{s,soil,min}$ | Minimum resistance soil evaporation | [s m$^{-1}$] | 50 |
| $g_D$ | Vapour pressure deficit correction factor for surface resistance | [-] | 0 |
| $h_{can}$ | Canopy height | [m] | GLAS (static) |
| $z_{0m}$ | Roughness length for momentum | [m] | 0.1 x $h_{can}$ (static) |
| $z_{0h}$ | Roughness length for heat and moisture | [m] | 0.1 x $z_{0m}$ (static) |
| α | Surface albedo | [-] | MOD44B (static) |
| $T_s$ | Initial surface temperature | [K] | ERA-Interim (3-hourly) |
| $T_{soil.1}$ | Temperature top soil layer | [K] | ERA-Interim (3-hourly) |
| $T_{soil,2}$ | Temperature deeper soil layer | [K] | ERA-Interim (3-hourly) |
| $\omega_{sat}$ | Saturated volumetric water content | [m$^3$ m$^{-3}$] | IGBP-DIS (static) |

**Soil**

| Symbol | Name | Unit | Default value or source |
|---|---|---|---|
| $\omega_{fc}$ | Volumetric water content field capacity | [m$^3$ m$^{-3}$] | IGBP-DIS (static) |
| $\omega_{wilt}$ | Volumetric water content wilting point | [m$^3$ m$^{-3}$] | IGBP-DIS (static) |
| EF | Evaporative fraction | [-] | ERA5 (daily) |
| $\omega_{soil,1}$ | Volumetric water content top soil layer | [m$^3$ m$^{-3}$] | By iterative matching of EF |
| $\omega_{soil,2}$ | Volumetric water content deeper soil layer idem | | *Idem* |
| $c_{veg}$ | Vegetation fraction | [-] | MOD44B (static) |
| a | Clapp and Hornberger retention curve parameter | [-] | HWSD (static) |
| b | Clapp and Hornberger retention curve parameter | [-] | HWSD (static) |
| p | Clapp and Hornberger retention curve parameter | [-] | HWSD (static) |
| $C_{Gsat}$ | Saturated soil conductivity for heat | [K m$^{-2}$ J$^{-1}$] | HWSD (static) |
| $C_{2,sat}$ | Coefficient force term moisture | [-] | HWSD (static) |
| $C_{2,ref}$ | Coefficient restore term moisture | [-] | HWSD (static) |
| Λ | Thermal diffusivity skin layer | [-] | 5.9 |

**Air**

| Symbol | Name | Unit | Default value or source |
|---|---|---|---|

| | | | |
|---|---|---|---|
| β | Ratio between buoyancy virtual heat and entrainment virtual heat | [-] | 0.2 |
| $\gamma_\theta$ | Initial lapse rate of potential temperature in the free atmosphere | [K m$^{-1}$] | From profile (IGRA) |
| $\gamma_q$ | Initial lapse rate of specific humidity in the free atmosphere | [kg kg$^{-1}$ m$^{-1}$] | *Idem* |
| $\gamma_u$ | Initial lapse rate of zonal wind in the free atmosphere | [s$^{-1}$] | *Idem* |
| $\gamma_v$ | Initial lapse rate of meridional wind in the free atmosphere | [s$^{-1}$] | *Idem* |
| $\Delta\theta_h$ | Initial temperature jump between the mixed layer and free atmosphere | [K] | *Idem* |
| $\Delta\theta_q$ | Initial specific humidity jump between the mixed layer and free atmosphere | [kg kg$^{-1}$] | *Idem* |
| $\Delta\theta_u$ | Initial specific zonal wind jump between the mixed layer and free atmosphere | [m s$^{-1}$] | *Idem* |
| $\Delta\theta_v$ | Initial specific meridional wind jump between the mixed layer and free atmosphere | [m s$^{-1}$] | *Idem* |

Table 2. Main input parameters for CLASS4GL. The parameter specifications and source acronyms are explained in section 2.3 and table 1.
"

*3. Figure 2: It took me a while to fi-nd the big X for the BLLAST experiment location, in part because much of the X is on top of country lines. Perhaps you can use a different Symbol.*

The symbol for the BLLAST location is now replaced with a thick cross. It will now appear as follows, which should make it more traceable (P7):

[Figure]

| | | |
|---|---|---|
| **Af** equatorial - fully humid | **Csa** warm temperate - summer dry - hot summer | **Dsb** snow - summer dry - warm summer |
| **Am** equatorial - monsoonal | **Csb** warm temperate - summer dry - warm summer | **Dsc** snow - summer dry - cool summer |
| **As** equatorial - summer dry | **Csc** warm temperate - summer dry - cool summer | **Dsd** snow - summer dry - extremely continental |
| **Aw** equatorial - winter dry | **Cwa** warm temperate - winter dry - hot summer | **Dwa** snow - winter dry - hot summer |
| **BSh** arid - steppe - hot arid | **Cwb** warm temperate - winter dry - warm summer | **Dwb** snow - winter dry - warm summer |
| **BSk** arid - steppe - cold arid | **Cwc** warm temperate - winter dry - cool summer | **Dwc** snow - winter dry - cool summer |
| **BWh** arid - desert - hot arid | **Dfa** snow - fully humid - hot summer | **Dwd** snow - winter dry - extremely continental |
| **BWk** arid - desert - cold arid | **Dfb** snow - fully humid - warm summer | **EF** polar - polar frost |
| **Cfa** warm temperate - fully humid - hot summer | **Dfc** snow - fully humid - cool summer | **ET** polar - polar tundra |
| **Cfb** warm temperate - fully humid - warm summer | **Dfd** snow - fully humid - extremely continental | **Ocean** |
| **Cfc** warm temperate - fully humid - cool summer | **Dsa** snow - summer dry - hot summer | ▲ HUMPPA ✖ BLLAST ✚ GOAMAZON |

*4. Section 3, line 3: When you first mention daytime tendencies, can you clarify what time period the resultant values are averaged over? I imagine it is from sunrise through the time of the second sounding.*

The tendencies are averaged from the morning sounding to the afternoon sounding. This will appear explicitly in the revised text as follows:

"The evaluation is done by comparing the modelled daytime tendencies of the mixed-layer height (*dh/dt*), potential temperature (*dθ/dt*) and specific humidity (*dq/dt*) against the corresponding tendencies observed from the balloon sounding pairs. **Observed and modelled tendencies represent the mean diurnal change from the morning sounding to the afternoon sounding.**" P14R4-7

*Since the local timing of this second sounding is at different times of day in different longitudes, might this introduce a spatial bias since BL growth rates are not uniform over the course of the day?*

Ideally, one would always have a sounding at the same local time in the morning and in the afternoon for every location. Since the launching times are based on UTC, this is obviously not the case. So we agree that the common local time of the sounding launch depends on longitude, and also that the ABL growth is certainly not uniform over the course of the day. However, the latter is taken into account, since the model is always initialized with the morning sounding for which the initial local model time is set equal to the sounding launch, and the same is true for the afternoon sounding. So it can be concluded that the expected tendency for each

launch or site (depending on the local time window being considered in the computation of that tendency) is equivalent for observations and model simulations, hence any biases related to launching times between the two are avoided.

This will be clarified with the following additional text (it will be located just after the previous text):

"**It should be noted that the local time of the morning and afternoon soundings changes given that the launch times are often at 0 and 12 h UTC, and that the boundary-layer tendencies are not uniform over the course of the day. The resulting variety in the tendencies is taken into account in the simulations, since the model is initialized with the morning sounding while the initial solar local time in the model is set equal to the sounding launch. The same happens for the end of the simulation at the time of the afternoon sounding. Therefore, the expected tendency for each launch or site (depending on the local time window being considered in the computation of that tendency) is equivalent for observations and models, hence any biases related to launching times between the two are avoided.**" P14R7-13

*5. Page 12, line 2: Here you mention the observed daytime tendencies when you are discussing the results from the three intensive campaigns. Are the results you show actually subdaily averages since you have more than two soundings per day?*

Yes, as clarified above, the results reflect the diurnal tendencies averaged over the time span between the morning and afternoon sounding, hence depend on the sounding launch times. In case there are multiple soundings retained during a particular day which is especially the case for the campaigns, the sounding closest to sunrise is taken for the initialization in the morning, and the latest sounding for the validation in the afternoon. This was not mentioned explicitly in the text. This clarification will now be added in the methodology section '2.2. Automized balloon data mining':

"... Here, the afternoon radiosonde profile on the same day needs to occur between local noon and 1 h before sunset **(defined as the time when the incoming shortwave radiation at the top of the atmosphere becomes zero)**, and at least 4 h after  **the model initialization by the morning sounding so that a sufficiently large model time span is considered. In case there are more than two soundings retained during a particular day which especially occurs during the campaigns, the sounding closest to sunrise is taken for the initialization in the morning, and the latest sounding for the validation in the afternoon**. so that a sufficiently large model time span is considered. In case there are more than two soundings retained during a particular day which especially occurs during the campaigns, the sounding closest to sunrise is taken for the initialization in the morning, and the latest sounding for the validation in the afternoon. ..." P9R24-35

*6. Figure 3: the correlation plots are quite busy, making it a touch hard to find the three symbols of interest. Maybe you can make the grey lines a little bit lighter grey so the symbols are easier to see.*

The grid lines are now lighter which increased the visibility:

[Figure]

The updated figure is now included in the revised manuscript (P15).

*7. Page 13, line 8: It is not clear to me where the 22% value comes from. Please clarify.*

The 22% value refers to the departure of the (normalized) standard deviation of the model output from the (normalized) standard deviation of the observations. This can be seen on the Taylor plots in Fig. 4 (figure pasted below) for which the centers of the open circles are between 0.78 and 1.22 of the normalized standard deviation. This will be made clear in the text as follows:

"In addition, the overall modelled range in dh/dt, dθ/dt and dq/dt agrees well with the observed range, with departures from the standard deviation of the observations below 22% – see Taylor plots in Fig. 4., **for which the departure of the modelled (normalized) standard deviation from the observed (normalized) standard deviation of each tendency is below 22%. This can be seen in the Taylor plots in Fig. 4 for which the centers of the open circles are between 0.78 and 1.22 of the normalized standard deviation.**" P16R12-16

[Figure]

*Reviewer 2*
*The paper presents a significant advancement in producing both useful accessible boundary layer data from radiosondes, and a nice marriage with a simple ABL model to produce continuous ABL data constrained by analyses, with open-source software.*

We would like to thank the referee for providing his/her review of the manuscript, and we are very glad regarding the appreciation of the software's potential. We also appreciate the comments, especially the suggestions related to possible sources of model biases. We provide a point-by-point answer below. Text of the revised manuscript is "quoted" in which the changes are provided in **bold** and removed text as . The location of text changes in the manuscript are provided in red. Unless specified otherwise, page and line numbers refer to the revised manuscript version with track changes.

*I was not able to fully run the CLASS4GL software myself. On a Mac using MacPorts, the PyYAML was not available and I downloaded directly from the website - there were issues*

*recognizing the CLoader option - apparently a version inconsistency. But I will follow through as I would like to use this tool.*

Thank you for testing software! The goal is to have a platform-independent software, so we strive to make it work on all platforms including Mac systems. As a solution on Mac, we would like to suggest to try either a Python environment with anaconda (as explained on https://class4gl.eu/?page_id=105) or Pycharm+homebrew. In case of pycharm+homebrew, these are the needed steps to install the CLoader module:

```
brew install libyaml-dev
pip install pyyaml
```

Please note that 'brew install libyaml' (so without '-dev') will not work. A similar solution may exist in case of your current Python environment using MacPorts. The CLoader is required to read yaml files 10–100 times faster, but depends on modules written in the C language.

*Regarding the manuscript, I suggest only minor changes are needed (editorial and regarding content), as outlined below:*

*P3 L14: Use "automates" instead of "automises". Likewise on P4 L7.*
*P4 L3: Change "dirunal" to "diurnal"*

Thanks for identifying the typos. They are corrected in the revised manuscript at P7R1, P18R2 and P4R3.

*P5, L10-11: It is a common assumption that the heat, moisture and momentum content of the ABL are perfectly mixed, but of course there will be mean vertical gradients, especially near the entrainment zone and the surface. In other words, the gradients here are a little weaker than for a well-mixed ABL, which may be compensated by other parameter choices. What would be the effect of specifying more realistic but still simple tails (e.g., exponential or even linear) of theta, q and V at the top and bottom of the ABL? This will relate to comments below regarding apparent biases.*

It is true that the ABL model considers a perfectly-mixed ABL with values of potential temperature, specific humidity and wind speed that are constant throughout the mixed layer, whereas the entrainment zone is represented as a jump between the mixed-layer values of and the free atmosphere values, and the surface layer as a analytic profile between mixed-layer values and the surface values. Other gradients within of the ABL are not explicitly represented. The Monin-Obukhov similarity theory is employed for calculating analytic surface layer profiles (and gradients) and the gradient transport in surface layer in an implicit way as a replacement for a more explicit representation. For the entrainment zone, the heat entrainment ratio (β) of 0.2 (the ratio of heat entrainment to heating through the surface layer) is considered and the additional entrainment by wind shear, based on observations and large eddy simulations (Vilà-Guerau de Arellano et al., 2015). More realistic tails at the top and bottom of the ABL are not yet considered in the ABL model, hence, it is not possible to quantify their effect and the possible associated biases in the model. Therefore, one would require a dedicated study for which one needs substantial changes to the ABL model formulations. We are aware about the

model limitations and associated uncertainties, and about the need for more research employing more realistic profiles. This is now mentioned more explicitly in the revised manuscript as follows:

"The use of the mixed-layer equations implies that turbulence **and vertical gradients** inside the mixed layer are not explicitly **re**solved, and the potential temperature (θ), specific humidity (q) and wind components are assumed to be homogeneous within the **mixed layer**. This assumption tends to be supported by the efficient turbulent mixing under convective conditions (Bauer, 1908). At the top of the **mixed layer**, the entrainment of heat and moisture is parameterized by a jump of θ, q and wind components over an infinitesimally small height, which  are initialized with a constant lapse rate with height in the overlying free atmosphere. Entrainment flux is calculated as a fixed fraction (0.2) of the buoyancy flux,  **to** which one also adds the entrainment flux driven by shear. An important feature of the model is the possibility to represent the subsidence coupled to the entrainment process at the inversion zone (Vilà-Guerau de Arellano et al., 2015). *[following text was moved up]* The surface–atmosphere turbulent exchanges for momentum, heat, and moisture **in the surface layer** are calculated considering their aerodynamic resistances. These are calculated in an iterative way assuming constant values for aerodynamic roughness lengths, while applying correction factors for non-neutral stratification of the atmospheric surface-layer (Paulson, 1970) according to the Monin–Obukhov similarity theory (Monin and Obukhov, 1954). **It should be kept in mind that more realistic profiles with explicit ABL gradients for temperature, humidity and wind speed – especially at the top (entrainment zone) and bottom (surface layer) of the mixed layer – are not yet considered by the model. In order to tackle these limitations and associated uncertainties, more research is needed employing more realistic profiles.**" P5R9-R25

P7 L4-10: Please state how many (or what percentage) of the 42,000 profiles are excluded for each reason (lacking both 00 and 12UTC soundings vs. non well-mixed profiles? The first seems a hard criterion, but exactly how well-mixed is that criterion and what if it is relaxed?

The criterion for a well-mixed profile is that the root mean square error of the profile measurements in the boundary layer is lower than 1.5°C. This information is now added to the manuscript. In addition, we provide the statistics on the reasons of profile retainment for each filtering step. Therefore, paragraph 2.2 is revised and it reads in the revised manuscript as follows (additional information is indicated in bold):

"2.2 Automized balloon data mining

[revised manuscript text omitted]

The above criteria are flexible and may be reconsidered according to the intended application, since there is an obvious trade-off between sounding quality and amount of data being retained. ..." P7R1-P10R8 [*end of the section can be found in the next reply below*]

*P7 L12: Change "says" to "days".*

Done. P10R9

*P7 L13-17: Are there clear discrepancies between the behavior and/or statistics of gap-filled (model) versus observationally driven results? I assume you have looked at this - a caveat might be warranted here.*

At this stage, the possibility of extracting profiles from continuous/gap-filled datasets (reanalysis, satellite-based products, and Earth system models...) is only implemented. The discrepancies between gap-filled versus observationally driven ABL model results have not been done yet. We agree that this is an important caveat, so the user should be warned that such a validation is needed as soon as a gap-filled dataset is employed. Hence, the following text will be added to the revised manuscript:

"It should be noted that many sites have only a few days with sounding pairs that meet the selection criteria, and most often, there are only intermittent time series available. For applications that require continuous datasets, the extraction profiles from reanalysis data, satellite-based products, or even Earth System Models, is also implemented in CLASS4GL. This alternative to the use of sounding data holds great promise for spatially-explicit climatological ABL studies and multi-annual trend assessments. **ABL model simulations using continuous/gap-filled datasets may deviate from those using the observations. Hence, an additional validation is needed as soon as such datasets are employed, in which one should compare the gap-filled datasets and the observations and the error propagation on the ABL model simulations**. Such an in-depth evaluation against the available sounding pairs can be done using the  present framework ." P10R9-16

*P10 L8: Change "reassure" to "assure".*

Done. P11R7

*Figure 3: There is only a circle (All) for dq/dt - not the other rates. Is something missing?*

They were indeed missing. All symbols will appear in the revised manuscript, as shown in the figure below (P15):

[Figure]

*P12 L6: Here I start wondering about the sources of biases and if you have been able to examine them. For dq/dt, a positive evaporation bias, excessive low-level moisture flux convergence (in the boundary conditions) or too little entrainment of dry air could each explain this. Has it been investigated? Is it likely a problem with the model or forcing data?*

It is true that biases are found in the model, which need to be tackled during the continuous development of CLASS4GL. We also expect that the bias have multiple origins, so both the ABL model (physical concepts) and its forcing data (convergence/advection, evaporation bias, cloud cover...) but also model tuning parameters (eg., entrainment ratio) and errors in the sounding observations used to initialize and validate the model, and all of these possible errors should be investigated in the further development of CLASS4GL. We will make this clear by adding the following text in the revised manuscript as follows:

"**The bias is expected to have multiple origins, including the ABL model and its physical concepts, the forcing data (convergence/advection, evaporation bias, cloud cover…, see Table 2), model tuning parameters (such as the entrainment ratio, see Table 2) and errors in the sounding observations used to initialize and validate the model. All these possible**

**error sources should be investigated in further development of CLASS4GL.**" P15R8-P16R1

The model performance statistics could already be when using one-hourly values (ERA5) instead of daily values (GLEAM) for evaporative fraction for the campaigns (see figure above) and the global results (see figure below, which is discussed further below). Particularly, the bias of *dq/dt* for the campaigns is now 0.04 g kg$^{-1}$ h$^{-1}$ (see figure above), whereas it was much higher in the previous results 0.17 g kg$^{-1}$ h$^{-1}$. The clear change in performance statistics by changing the *EF* supports that the sources of biases are partly originating from errors propagating from the forcing data.

Since ERA5 provides better performance statistics, this is now used for *EF* in the results section, which is mentioned in the manuscript:

"However for the results presented in Section 3, one-hourly values of *EF* from recently released reanalysis dataset ERA5 (Copernicus Climate Change Service (C3S), 2017) are used instead of daily values from GLEAM, which provided a higher temporal resolution and better performance statistics." P11R11-13

The new results are also included in the revised manuscript. Using the updated results as shown with the figures above, we have modified the text accordingly:

"However, we could identify common model limitations over the three campaigns, see Fig. 3. . **This includes an overall (slight) positive bias in dh/dt (dθ/dt and dq/dt), and an under(over)estimation of its (their) variability** as indicated with **a** normalized standard deviation **different from**  1 in the Taylor plots (Fig. 3)**.**" P15R5-R8

*Fig 4 and associated text: If the heating and moistening rates are converted to J/kg/h by multiplying by C_p and lambda_v respectively, we get that the heating bias is -52 J/kg/h but the positive moistening bias is 175 J/kg/h.*

Using the one-hourly ERA5 1-hourly data as forcing for evaporative fraction as discussed above, we could also obtain overall better model performance statistics for the IGRA global soundings (see figure below) than with GLEAM daily input: particularly, the global negative bias in dθ/dt has been reduced from -0.052K/h (-52 J kg$^{-1}$ h$^{-1}$) to -0.036K h$^{-1}$ (-36 J kg$^{-1}$ h$^{-1}$), and the positive bias in dq/dt has been reduced from 0.07 g kg$^{-1}$ h$^{-1}$ (or 175 J kg$^{-1}$ h$^{-1}$) to 0.06 g kg$^{-1}$ h$^{-1}$ (or 150 J kg$^{-1}$ h$^{-1}$). Pearson correlation coefficients have slightly improved (0.54/0.79/0.52 -> 0.53/0.82/0.54), and all RMSEs are slightly lower as well (72.3 m h$^{-1}$ / 0.211 K h$^{-1}$ / 0.17 g kg$^{-1}$ h$^{-1}$ -> 68.8m h$^{-1}$ / 0.202 K h$^{-1}$ / 0.16 g kg$^{-1}$ h$^{-1}$).

[Figure]

These results are now updated in the revised manuscript, and the text and discussions will be modified accordingly:

"The simulated diurnal ABL tendencies show a similar accuracy when evaluated against the global IGRA sounding archive, despite the fact that these operational sounding have a priori lower quality standards; this applies to all three ABL tendencies shown in Fig. 4. Here, the  **magnitude of the bias is of the same order** for the change in θ ( **-0.036** K h⁻¹)  **as** for the intense observation campaigns,  **but is now negative. The biases** are smaller for *h* ( 10.1 m h⁻¹) compared to the campaigns and slightly larger for q (**0.06** g kg⁻¹). As for the research campaigns, the model is able to reproduce the variability among the different operational sounding days, with Pearson correlation coefficients of  **0.53, 0.82 and 0.54** for the diurnal tendencies of h, θ and q, respectively." P16R6-12

*The discrepancy is 123 J/kg/h – ...*

With the updated results mentioned above, the discrepancy has now been slightly reduced from 123 J kg⁻¹ h⁻¹ to 114 J kg⁻¹ h⁻¹.

*… again there could be multiple sources of this. First thought is net radiation, but excessive ground heat flux from the soil, advection (convergence) or entrainment could all be reasons. Any idea about the source of this net energy bias?*

*P13 L25-29: Related to above, a nice speculation on causes, but atmospheric models including reanalyses tend to have too much surface net radiation due to cloud errors and lack or proper representation of aerosol effects. R_Net or the input ERA-I radiation should be validated against independent data (e.g., the available CERES data) as a sanity check.*

In line with the discussion above, we expect that the source of the net energy bias have multiple origins, including the ABL model and its physical concepts, the forcing data, model tuning parameters and errors in the sounding observations. We agree with the suggestions of the referee that this could be especially due to 
[revised manuscript text omitted]